# META-LEARNING WITHOUT MEMORIZATION

**Mingzhang Yin**[12], **George Tucker**[2], **Mingyuan Zhou**[1], **Sergey Levine**[23], **Chelsea Finn**[24]
mzyin@utexas.edu, gjt@google.com, mingyuan.zhou@mccombs.utexas.edu
svlevine@eecs.berkeley.edu, cbfinn@cs.stanford.edu
[1]UT Austin, [2]Google Research, Brain team, [3]UC Berkeley, [4]Stanford

## ABSTRACT

The ability to learn new concepts with small amounts of data is a critical aspect of intelligence that has proven challenging for deep learning methods. Meta-learning has emerged as a promising technique for leveraging data from previous tasks to enable efficient learning of new tasks. However, most meta-learning algorithms implicitly require that the meta-training tasks be *mutually-exclusive*, such that no single model can solve all of the tasks at once. For example, when creating tasks for few-shot image classification, prior work uses a per-task random assignment of image classes to N-way classification labels. If this is not done, the meta-learner can ignore the task training data and learn a single model that performs all of the meta-training tasks zero-shot, but does not adapt effectively to new image classes. This requirement means that the user must take great care in designing the tasks, for example by shuffling labels or removing task identifying information from the inputs. In some domains, this makes meta-learning entirely inapplicable. In this paper, we address this challenge by designing a meta-regularization objective using information theory that places precedence on data-driven adaptation. This causes the meta-learner to decide what must be learned from the task training data and what should be inferred from the task testing input. By doing so, our algorithm can successfully use data from *non-mutually-exclusive* tasks to efficiently adapt to novel tasks. We demonstrate its applicability to both contextual and gradient-based meta-learning algorithms, and apply it in practical settings where applying standard meta-learning has been difficult. Our approach substantially outperforms standard meta-learning algorithms in these settings.

## 1 INTRODUCTION

The ability to learn new concepts and skills with small amounts of data is a critical aspect of intelligence that many machine learning systems lack. Meta-learning (Schmidhuber, 1987) has emerged as a promising approach for enabling systems to quickly learn new tasks by building upon experience from previous related tasks (Thrun & Pratt, 2012; Koch et al., 2015; Santoro et al., 2016; Ravi & Larochelle, 2016; Finn et al., 2017). Meta-learning accomplishes this by explicitly optimizing for few-shot generalization across a set of meta-training tasks. The meta-learner is trained such that, after being presented with a small task training set, it can accurately make predictions on test datapoints for that meta-training task.

While these methods have shown promising results, current methods require careful design of the meta-training tasks to prevent a subtle form of *task overfitting*, distinct from standard overfitting in supervised learning. If the task can be accurately inferred from the test input alone, then the task training data can be ignored while still achieving low meta-training loss. In effect, the model will collapse to one that makes zero-shot decisions. This presents an opportunity for overfitting where the meta-learner generalizes on meta-training tasks, but fails to adapt when presented with training data from novel tasks. We call this form of overfitting the *memorization problem* in meta-learning because the meta-learner memorizes a function that solves all of the meta-training tasks, rather than learning to adapt.

Existing meta-learning algorithms implicitly resolve this problem by carefully designing the meta-training tasks such that no single model can solve all tasks zero-shot; we call tasks constructed in this

---

Implementation and examples available here: `https://github.com/google-research/google-research/tree/master/meta_learning_without_memorization`.

way *mutually-exclusive*. For example, for $N$-way classification, each task consists of examples from $N$ randomly sampled classes. The $N$ classes are labeled from 1 to $N$, and critically, for each task, we *randomize* the assignment of classes to labels $\{1, 2, \ldots, N\}$ (visualized in Appendix Figure 3). This ensures that the task-specific class-to-label assignment cannot be inferred from a test input alone. However, the mutually-exclusive tasks requirement places a substantial burden on the user to cleverly design the meta-training setup (e.g., by shuffling labels or omitting goal information). While shuffling labels provides a reasonable mechanism to force tasks to be mutually-exclusive with standard few-shot image classification datasets such as MiniImageNet (Ravi & Larochelle, 2016), this solution cannot be applied to all domains where we would like to utilize meta-learning. For example, consider meta-learning a pose predictor that can adapt to different objects: even if $N$ different objects are used for meta-training, a powerful model can simply learn to ignore the training set for each task, and directly learn to predict the pose of each of the $N$ objects. However, such a model would not be able to adapt to *new* objects at meta-test time.

The primary contributions of this work are: 1) to identify and formalize the memorization problem in meta-learning, and 2) to propose a meta-regularizer (MR) using information theory as a general approach for mitigating this problem *without* placing restrictions on the task distribution. We formally differentiate the meta-learning memorization problem from overfitting problem in conventional supervised learning, and empirically show that naïve applications of standard regularization techniques do not solve the memorization problem in meta-learning. The key insight of our meta-regularization approach is that the model acquired when memorizing tasks is more complex than the model that results from task-specific adaptation because the memorization model is a single model that simultaneously performs well on all tasks. It needs to contain all information in its weights needed to do well on test points without looking at training points. Therefore we would expect the information content of the weights of a memorization model to be larger, and hence the model should be more complex. As a result, we propose an objective that regularizes the information complexity of the meta-learned function class (motivated by Alemi et al. (2016); Achille & Soatto (2018)). Furthermore, we show that meta-regularization in MAML can be rigorously motivated by a PAC-Bayes bound on generalization. In a series of experiments on non-mutually-exclusive task distributions entailing both few-shot regression and classification, we find that memorization poses a significant challenge for both gradient-based (Finn et al., 2017) and contextual (Garnelo et al., 2018a) meta-learning methods, resulting in near random performance on test tasks in some cases. Our meta-regularization approach enables both of these methods to achieve efficient adaptation and generalization, leading to substantial performance gains across the board on non-mutually-exclusive tasks.

## 2 PRELIMINARIES

We focus on the standard supervised meta-learning problem (see, e.g., Finn et al. (2017)). Briefly, we assume tasks $\mathcal{T}_i$ are sampled from a task distribution $p(\mathcal{T})$. During meta-training, for each task, we observe a set of training data $\mathcal{D}_i = (\boldsymbol{x}_i, \boldsymbol{y}_i)$ and a set of test data $\mathcal{D}_i^* = (\boldsymbol{x}_i^*, \boldsymbol{y}_i^*)$ with $\boldsymbol{x}_i = (x_{i1}, \ldots, x_{iK}), \boldsymbol{y}_i = (y_{i1}, \ldots, y_{iK})$ sampled from $p(x, y | \mathcal{T}_i)$, and similarly for $\mathcal{D}_i^*$. We denote the entire meta-training set as $\mathcal{M} = \{\mathcal{D}_i, \mathcal{D}_i^*\}_{i=1}^N$. The goal of meta-training is to learn a model for a new task $\mathcal{T}$ by leveraging what is learned during meta-training and a small amount of training data for the new task $\mathcal{D}$. We use $\theta$ to denote the meta-parameters learned during meta-training and use $\phi$ to denote the task-specific parameters that are computed based on the task training data.

Following Grant et al. (2018); Gordon et al. (2018), given a meta-training set $\mathcal{M}$, we consider meta-learning algorithms that maximize conditional likelihood $q(\hat{y}^* = y^* | x^*, \theta, \mathcal{D})$, which is composed of three distributions: $q(\theta | \mathcal{M})$ that summarizes meta-training data into a distribution on meta-parameters, $q(\phi | \mathcal{D}, \theta)$ that summarizes the per-task training set into a distribution on task-specific parameters, and $q(\hat{y}^* | x^*, \phi, \theta)$ that is the predictive distribution. These distributions are learned to minimize

$$-\frac{1}{N} \sum_i \mathbb{E}_{q(\theta|\mathcal{M})q(\phi|\mathcal{D}_i,\theta)} \left[ \frac{1}{K} \sum_{(x^*,y^*) \in \mathcal{D}_i^*} \log q(\hat{y}^* = y^* | x^*, \phi, \theta) \right]. \tag{1}$$

For example, in MAML (Finn et al., 2017), $\theta$ and $\phi$ are the weights of a predictor network, $q(\theta | \mathcal{M})$ is a delta function learned over the meta-training data, $q(\phi | \mathcal{D}, \theta)$ is a delta function centered at a point defined by gradient optimization, and $\phi$ parameterizes the predictor network $q(\hat{y}^* | x^*, \phi)$ (Grant et al., 2018). In particular, to determine the task-specific parameters $\phi$, the task training data $\mathcal{D}$ and $\theta$ are used in the predictor model $\phi = \theta + \frac{\alpha}{K} \sum_{(x,y) \in \mathcal{D}} \nabla_\theta \log q(y | x, \phi = \theta)$.

Another family of meta-learning algorithms are contextual methods (Santoro et al., 2016), such as conditional neural processes (CNP) (Garnelo et al., 2018b;a). CNP instead defines $q(\phi|\mathcal{D}, \theta)$ as a mapping from $\mathcal{D}$ to a summary statistic $\phi$ (parameterized by $\theta$). In particular, $\phi = a_\theta \circ h_\theta(\mathcal{D})$ is the output of an aggregator $a_\theta(\cdot)$ applied to features $h_\theta(\mathcal{D})$ extracted from the task training data. Then $\theta$ parameterizes a predictor network that takes $\phi$ and $x^*$ as input and produces a predictive distribution $q(\hat{y}^*|x^*, \phi, \theta)$.

In the following sections, we describe a common pitfall for a variety of meta-learning algorithms, including MAML and CNP, and a general meta-regularization approach to prevent this pitfall.

## 3  THE MEMORIZATION PROBLEM IN META-LEARNING

The ideal meta-learning algorithm will learn in such a way that generalizes to novel tasks. However, we find that unless tasks are carefully designed, current meta-learning algorithms can overfit to the tasks and end up ignoring the task training data (i.e., either $q(\phi|\mathcal{D}, \theta)$ does not depend on $\mathcal{D}$ or $q(\hat{y}^*|x^*, \phi, \theta)$ does not depend on $\phi$, as shown in Figure 1), which can lead to poor generalization. This memorization phenomenon is best understood through examples.

Consider a 3D object pose prediction problem (illustrated in Figure 1), where each object has a fixed canonical pose. The $(x, y)$ pairs for the task are 2D grey-scale images of the rotated object $(x)$ and the rotation angle relative to the fixed canonical pose for that object $(y)$. In the most extreme case, for an unseen object, the task is impossible without using $\mathcal{D}$ because the canonical pose for the unseen object is unknown. The number of objects in the meta-training dataset is small, so it is straightforward for a single network to memorize the canonical pose for each training object and to infer the object from the input image (i.e., task overfitting), thus achieving a low training error without using $\mathcal{D}$. However, by construction, this solution will necessarily have poor generalization to test tasks with unseen objects.

As another example, imagine an automated medical prescription system that suggests medication prescriptions to doctors based on patient symptoms and the patient's previous record of prescription responses (i.e., medical history) for adaptation. In the meta-learning framework, each patient represents a separate task. Here, the symptoms and prescriptions have a close relationship, so we *cannot* assign random prescriptions to symptoms, in contrast to the classification tasks where we *can* randomly shuffle the labels to create mutually-exclusiveness. For this non-mutually-exclusive task distribution, a standard meta-learning system can memorize the patients' identity information in the training, leading it to ignore the medical history and only utilize the symptoms combined with the memorized information. As a result, it may issue highly accurate prescriptions on the *meta-training* set, but fail to adapt to new patients effectively. While such a system would achieve a baseline level of accuracy for new patients, it would be no better than a standard supervised learning method applied to the pooled data.

We formally define (complete) memorization as:

**Definition 1** (Complete Meta-Learning Memorization). *Complete memorization in meta-learning is when the learned model ignores the task training data such that $I(\hat{y}^*; \mathcal{D}|x^*, \theta) = 0$ (i.e., $q(\hat{y}^*|x^*, \theta, \mathcal{D}) = q(\hat{y}^*|x^*, \theta) = \mathbb{E}_{\mathcal{D}'|x^*}[q(\hat{y}^*|x^*, \theta, \mathcal{D}')]$).*

Memorization describes an issue with overfitting the meta-training tasks, but it does not preclude the network from generalizing to unseen $(x, y)$ pairs on the tasks similar to the training tasks. Memorization becomes an undesired problem for generalization to new tasks when $I(y^*; \mathcal{D}|x^*) \gg I(\hat{y}^*; \mathcal{D}|x^*, \theta)$ (i.e., the task training data is necessary to achieve good performance, even with exact inference under the data generating distribution, to make accurate predictions).

A model with the memorization problem may generalize to new datapoints in training tasks but cannot generalize to novel tasks, which distinguishes it from typical overfitting in supervised learning. In practice, we find that MAML and CNP frequently converge to this memorization solution (Table 2). For MAML, memorization can occur when a particular setting of $\theta$ that does not adapt to the task training data can achieve comparable meta-training error to a solution that adapts $\theta$. For example, if a setting of $\theta$ can solve all of the meta-training tasks (i.e., for all $(x, y)$ in $\mathcal{D}$ and $\mathcal{D}^*$ the predictive error is close to zero), the optimization may converge to a stationary point of the MAML objective where minimal adaptation occurs based on the task training set (i.e., $\phi \approx \theta$). For a novel task where it is necessary to use the task training data, MAML can in principle still leverage the task training data because the adaptation step is based on gradient descent. However, in practice, the

poor initialization of $\theta$ can affect the model's ability to generalize from a small mount of data. For CNP, memorization can occur when the predictive distribution network $q(\hat{y}^*|x^*, \phi, \theta)$ can achieve low training error without using the task training summary statistics $\phi$. On a novel task, the network is not trained to use $\phi$, so it is unable to use the information extracted from the task training set to effectively generalize.

In some problem domains, the memorization problem can be avoided by carefully constructing the tasks. For example, for $N$-way classification, each task consists of examples from $N$ randomly sampled classes. If the classes are assigned to a random permutation of $N$ for each task, this ensures that the task-specific class-to-label assignment cannot be inferred from the test inputs alone. As a result, a model that ignores the task training data cannot achieve low training error, preventing convergence to the memorization problem. We refer to tasks constructed in this way as *mutually-exclusive*. However, the mutually-exclusive tasks requirement places a substantial burden on the user to cleverly design the meta-training setup (e.g., by shuffling labels or omitting goal information) and cannot be applied to all domains where we would like to utilize meta-learning.

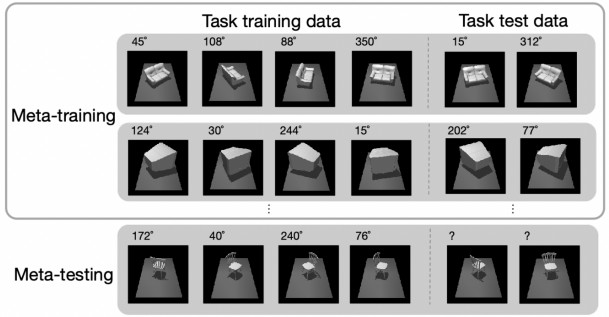 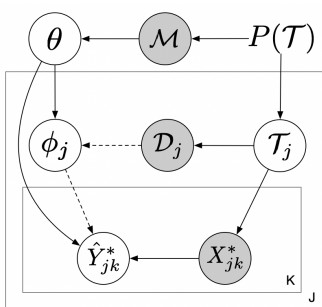

Figure 1: Left: An example of non-mutually-exclusive pose prediction tasks, which may lead to the memorization problem. The training tasks are non-mutually-exclusive because the test data label (right) can be inferred accurately without using task training data (left) in the training tasks, by memorizing the canonical orientation of the meta-training objects. For a new object and canonical orientation (bottom), the task cannot be solved without using task training data (bottom left) to infer the canonical orientation. Right: Graphical model for meta-learning. Observed variables are shaded. Without either one of the dashed arrows, $\hat{Y}^*$ is conditionally independent of $\mathcal{D}$ given $\theta$ and $X^*$, which we refer to as complete memorization (Definition 1).

## 4 META REGULARIZATION USING INFORMATION THEORY

At a high level, the sources of information in the predictive distribution $q(\hat{y}^*|x^*, \theta, \mathcal{D})$ come from the input, the meta-parameters, and the data. The memorization problem occurs when the model encodes task information in the predictive network that is readily available from the task training set (i.e., it memorizes the task information for each meta-training task). We could resolve this problem by encouraging the model to minimize the training error and to rely on the task training dataset as much as possible for the prediction of $y^*$ (i.e., to maximize $I(\hat{y}^*; \mathcal{D}|x^*, \theta)$). Explicitly maximizing $I(\hat{y}^*; \mathcal{D}|x^*, \theta)$ requires an intractable marginalization over task training sets to compute $q(\hat{y}^*|x^*, \theta)$. Instead, we can implicitly encourage it by restricting the information flow from other sources ($x^*$ and $\theta$) to $\hat{y}^*$. To achieve both low error and low mutual information between $\hat{y}^*$ and ($x^*, \theta$), the model must use task training data $\mathcal{D}$ to make predictions, hence increasing the mutual information $I(\hat{y}^*; \mathcal{D}|x^*, \theta)$, leading to reduced memorization. In this section, we describe two tractable ways to achieve this.

### 4.1 META REGULARIZATION ON ACTIVATIONS

Given $\theta$, the statistical dependency between $x^*$ and $\hat{y}^*$ is controlled by the direct path from $x^*$ to $\hat{y}^*$ and the indirect path through $\mathcal{D}$ (see Figure 1), where the latter is desirable because it leverages the task training data. We can control the information flow between $x^*$ and $\hat{y}^*$ by introducing an intermediate stochastic bottleneck variable $z^*$ such that $q(\hat{y}^*|x^*, \phi, \theta) = \int q(\hat{y}^*|z^*, \phi, \theta)q(z^*|x^*, \theta)\, dz^*$ (Alemi et al., 2016) as shown in Figure 4. Now, we would like

to maximize $I(\hat{y}^*; \mathcal{D}|z^*, \theta)$ to prevent memorization. We can bound this mutual information by

$$
\begin{aligned}
&I(\hat{y}^*; \mathcal{D}|z^*, \theta) \\
&\geq I(x^*; \hat{y}^*|\theta, z^*) = I(x^*; \hat{y}^*|\theta) - I(x^*; z^*|\theta) + I(x^*; z^*|\hat{y}^*, \theta) \\
&\geq I(x^*; \hat{y}^*|\theta) - I(x^*; z^*|\theta) \\
&= I(x^*; \hat{y}^*|\theta) - \mathbb{E}_{p(x^*)q(z^*|x^*, \theta)} \left[ \log \frac{q(z^*|x^*, \theta)}{q(z^*|\theta)} \right] \\
&\geq I(x^*; \hat{y}^*|\theta) - \mathbb{E} \left[ \log \frac{q(z^*|x^*, \theta)}{r(z^*)} \right] = I(x^*; \hat{y}^*|\theta) - \mathbb{E} \left[ D_{\text{KL}}(q(z^*|x^*, \theta)||r(z^*)) \right]
\end{aligned}
\tag{2}
$$

where $r(z^*)$ is a variational approximation to the marginal, the first inequality follows from the statistical dependencies in our model (see Figure 4 and Appendix A.2 for the proof). By simultaneously minimizing $\mathbb{E} \left[ D_{\text{KL}}(q(z^*|x^*, \theta)||r(z^*)) \right]$ and maximizing the mutual information $I(x^*; \hat{y}^*|\theta)$, we can implicitly encourage the model to use the task training data $\mathcal{D}$.

For non-mutually-exclusive problems, the true label $y^*$ is dependent on $x^*$. If the model has the memorization problem and $I(x^*; \hat{y}^*|\theta) = 0$, then $q(\hat{y}^*|x^*, \theta, \mathcal{D}) = q(\hat{y}^*|x^*, \theta) = q(\hat{y}^*|\theta)$, which means the model predictions do not depend on $x^*$ or $\mathcal{D}$. Hence, in practical problems, the predictions generated from the model will have low accuracy.

This suggests minimizing the training loss in Eq. (1) can increase $I(\hat{y}^*; \mathcal{D}|x^*, \theta)$ or $I(x^*; \hat{y}^*|\theta)$. Replacing the maximization of $I(x^*; \hat{y}^*|\theta)$ in Eq. (2) with minimizing the training loss results in the following regularized training objective

$$
\frac{1}{N} \sum_i \mathbb{E}_{q(\theta|\mathcal{M})q(\phi|\mathcal{D}_i, \theta)} \left[ -\frac{1}{K} \sum_{(x^*, y^*) \in \mathcal{D}_i^*} \log q(\hat{y}^* = y^*|x^*, \phi, \theta) + \beta D_{\text{KL}}(q(z^*|x^*, \theta)||r(z^*)) \right]
\tag{3}
$$

where $\log q(\hat{y}^*|x^*, \phi, \theta)$ is estimated by $\log q(\hat{y}^*|z^*, \phi, \theta)$ with $z^* \sim q(z^*|x^*, \theta)$, $\beta$ modulates the regularizer and $r(z^*)$ can be set as $\mathcal{N}(z^*; 0, I)$. We refer to this regularizer as meta-regularization (MR) on the activations.

As we demonstrate in Section 6, we find that this regularizer performs well, but in some cases can fail to prevent the memorization problem. Our hypothesis is that in these cases, the network can sidestep the information constraint by storing the prediction of $y^*$ in a part of $z^*$, which incurs a small penalty in Eq. (3) and small lower bound in Eq. (2).

## 4.2 META REGULARIZATION ON WEIGHTS

Alternatively, we can penalize the task information stored in the meta-parameters $\theta$. Here, we provide an informal argument and provide the complete argument in Appendix A.3. Analogous to the supervised setting (Achille & Soatto, 2018), given meta-training dataset $\mathcal{M}$, we consider $\theta$ as random variable where the randomness can be introduced by training stochasticity. We model the stochasticity over $\theta$ with a Gaussian distribution $\mathcal{N}(\theta; \theta_\mu, \theta_\sigma)$ with learned mean and variance parameters per dimension (Blundell et al., 2015; Achille & Soatto, 2018). By penalizing $I(y_{1:N}^*, \mathcal{D}_{1:N}; \theta|x_{1:N}^*)$, we can limit the information about the training tasks stored in the meta-parameters $\theta$ and thus require the network to use the task training data to make accurate predictions. We can tractably upper bound it by

$$
I(y_{1:N}^*, \mathcal{D}_{1:N}; \theta|x_{1:N}^*) = \mathbb{E} \left[ \log \frac{q(\theta|\mathcal{M})}{q(\theta|x_{1:N}^*)} \right] \leq \mathbb{E} \left[ D_{\text{KL}}(q(\theta|\mathcal{M})||r(\theta)) \right],
\tag{4}
$$

where $r(\theta)$ is a variational approximation to the marginal, which we set to $\mathcal{N}(\theta; 0, I)$. In practice, we apply meta-regularization to the meta-parameters $\theta$ that are not used to adapt to the task training data and denote the other parameters as $\tilde{\theta}$. In this way, we control the complexity of the network that can predict the test labels without using task training data, but we do not limit the complexity of the network that processes the task training data. Our final meta-regularized objective can be written as

$$
\frac{1}{N} \sum_i \mathbb{E}_{q(\theta; \theta_\mu, \theta_\sigma)q(\phi|\mathcal{D}_i, \tilde{\theta})} \left[ -\frac{1}{K} \sum_{(x^*, y^*) \in \mathcal{D}_i^*} \log q(\hat{y}^* = y^*|x^*, \phi, \theta, \tilde{\theta}) + \beta D_{\text{KL}}(q(\theta; \theta_\mu, \theta_\sigma)||r(\theta)) \right]
\tag{5}
$$

For MAML, we apply meta-regularization to the parameters uninvolved in the task adaptation. For CNP, we apply meta-regularization to the encoder parameters. The detailed algorithms are shown in Algorithm 1 and 2 in the appendix.

### 4.3 Does Meta Regularization Lead to Better Generalization?

Now that we have derived meta regularization approaches for mitigating the memorization problem, we theoretically analyze whether meta regularization leads to better generalization via a PAC-Bayes bound. In particular, we study meta regularization (MR) on the weights (W) of MAML, i.e. MR-MAML (W), as a case study.

Meta regularization on the weights of MAML uses a Gaussian distribution $\mathcal{N}(\theta; \theta_\mu, \theta_\sigma)$ to model the stochasticity in the weights. Given a task and task training data, the expected error is given by

$$er(\theta_\mu, \theta_\sigma, \mathcal{D}, \mathcal{T}) = \mathbb{E}_{\theta \sim \mathcal{N}(\theta; \theta_\mu, \theta_\sigma), \phi \sim q(\phi|\theta, \mathcal{D}), (x^*, y^*) \sim p(x, y|\mathcal{T})} \left[ \mathcal{L}(x^*, y^*, \phi) \right], \tag{6}$$

where the prediction loss $\mathcal{L}(x^*, y^*, \phi_i)$ is bounded[1]. Then, we would like to minimize the error on novel tasks

$$er(\theta_\mu, \theta_\sigma) = \mathbb{E}_{\mathcal{T} \sim p(\mathcal{T}), \mathcal{D} \sim p(x, y|\mathcal{T})} \left[ er(\theta_\mu, \theta_\sigma, \mathcal{D}, \mathcal{T}) \right] \tag{7}$$

We only have a finite sample of training tasks, so computing $er(Q)$ is intractable, but we can form an empirical estimate:

$$\hat{er}(\theta_\mu, \theta_\sigma, \mathcal{D}_1, \mathcal{D}_1^*, ..., \mathcal{D}_n, \mathcal{D}_n^*)$$

$$= \frac{1}{n} \sum_{i=1}^{n} \underbrace{\mathbb{E}_{\theta \sim \mathcal{N}(\theta; \theta_\mu, \theta_\sigma), \phi_i \sim q(\phi|\theta, \mathcal{D}_i)} \left[ -\frac{1}{K} \sum_{(x^*, y^*) \in \mathcal{D}_i^*} \log q(\hat{y}^* = y^* | x^*, \phi_i) \right]}_{\hat{er}(\theta_\mu, \theta_\sigma, \mathcal{D}_i, \mathcal{D}_i^*)} \tag{8}$$

where for exposition we have assumed $|\mathcal{D}_i^*| = K$ are the same for all tasks. We would like to relate $er(\theta_\mu, \theta_\sigma)$ and $\hat{er}(\theta_\mu, \theta_\sigma, \mathcal{D}_1, \mathcal{D}_1^*, ..., \mathcal{D}_n, \mathcal{D}_n^*)$, but the challenge is that $\theta_\mu$ and $\theta_\sigma$ are derived from the meta-training tasks $\mathcal{D}_1, \mathcal{D}_1^*, ..., \mathcal{D}_n, \mathcal{D}_n^*$. There are two sources of generalization error: (i) error due to the finite number of observed tasks and (ii) error due to the finite number of examples observed per task. Closely following the arguments in (Amit & Meir, 2018), we apply a standard PAC-Bayes bound to each of these and combine the results with a union bound, resulting in the following Theorem.

**Theorem 1.** *Let $P(\theta)$ be an arbitrary prior distribution over $\theta$ that does not depend on the meta-training data. Then for any $\delta \in (0, 1]$, with probability at least $1 - \delta$, the following inequality holds uniformly for all choices of $\theta_\mu$ and $\theta_\sigma$,*

$$er(\theta_\mu, \theta_\sigma) \leq \frac{1}{n} \sum_{i=1}^{n} \hat{er}(\theta_\mu, \theta_\sigma, \mathcal{D}_i, \mathcal{D}_i^*) +$$

$$\left( \sqrt{\frac{1}{2(K-1)}} + \sqrt{\frac{1}{2(n-1)}} \right) \sqrt{D_{KL}(\mathcal{N}(\theta; \theta_\mu, \theta_\sigma) \| P) + \log \frac{n(K+1)}{\delta}}, \tag{9}$$

*where $n$ is the number of meta-training tasks and $K$ is the number of per-task validation datapoints.*

We defer the proof to the Appendix A.4. The key difference from the result in (Amit & Meir, 2018) is that we leverage the fact that the task training data is split into training and validation.

In practice, we set $P(\theta) = r(\theta) = \mathcal{N}(\theta; 0, I)$. If we can achieve a low value for the bound, then with high probability, our test error will also be low. As shown in the Appendix A.4, by a first order Taylor expansion of the the second term of the RHS in Eq.(9) and setting the coefficient of the KL term as $\beta = \frac{\sqrt{1/2(K-1)} + \sqrt{1/2(n-1)}}{2\sqrt{\log n(K+1)/\delta}}$, we recover the MR-MAML(W) objective (Eq.(5)). $\beta$ trades-off between the tightness of the generalization bound and the probability that it holds true. The result of this bound suggests that the proposed meta-regularization on weights does indeed improve generalization on the meta-test set.

---

[1]In practice, $\mathcal{L}(x^*, y^*, \phi_i)$ is MSE on a bounded target space or classification accuracy. We optimize the negative log-likelihood as a bound on the 0-1 loss.

## 5 RELATED WORK

Previous works have developed approaches for mitigating various forms of overfitting in meta-learning. These approaches aim to improve generalization in several ways: by reducing the number of parameters that are adapted in MAML (Zintgraf et al., 2019), by compressing the task embedding (Lee et al., 2019), through data augmentation from a GAN (Zhang et al., 2018), by using an auxiliary objective on task gradients (Guiroy et al., 2019), and via an entropy regularization objective (Jamal & Qi, 2019). These methods all focus on the setting with mutually-exclusive task distributions. We instead recognize and formalize the memorization problem, a particular form of overfitting that manifests itself with non-mutually-exclusive tasks, and offer a general and principled solution. Unlike prior methods, our approach is applicable to both contextual and gradient-based meta-learning methods. We additionally validate that prior regularization approaches, namely TAML (Jamal & Qi, 2019), are not effective for addressing this problem setting.

Our derivation uses a Bayesian interpretation of meta-learning (Tenenbaum, 1999; Fei-Fei et al., 2003; Edwards & Storkey, 2016; Grant et al., 2018; Gordon et al., 2018; Finn et al., 2018; Kim et al., 2018; Harrison et al., 2018). Some Bayesian meta-learning approaches place a distributional loss on the inferred task variables to constrain them to a prior distribution (Garnelo et al., 2018b; Gordon et al., 2018; Rakelly et al., 2019), which amounts to an information bottleneck on the latent *task variables*. Similarly Zintgraf et al. (2019); Lee et al. (2019); Guiroy et al. (2019) aim to produce simpler or more compressed task adaptation processes. Our approach does the opposite, penalizing information from the *inputs* and *parameters*, to encourage the task-specific variables to contain greater information driven by the per-task data.

We use PAC-Bayes theory to study the generalization error of meta-learning and meta-regularization. Pentina & Lampert (2014) extends the single task PAC-Bayes bound (McAllester, 1999) to the multi-task setting, which quantifies the gap between empirical error on training tasks and the expected error on new tasks. More recent research shows that, with tightened generalization bounds as the training objective, the algorithms can reduce the test error for mutually-exclusive tasks (Galanti et al., 2016; Amit & Meir, 2018). Our analysis is different from these prior works in that we only include pre-update meta parameters in the generalization bound rather than both pre-update and post-update parameters. In the derivation, we also explicitly consider the splitting of data into the task training set and task validation set, which is aligned with the practical setting.

The memorization problem differs from overfitting in conventional supervised learning in several aspects. First, memorization occurs at the task level rather than datapoint level and the model memorizes functions rather than labels. In particular, within a training task, the model can generalize to new datapoints, but it fails to generalize to new tasks. Second, the source of information for achieving generalization is different. For meta-learning the information is from both the meta-training data and new task training data but in standard supervised setting the information is only from training data. Finally, the aim of regularization is different. In the conventional supervised setting, regularization methods such as weight decay (Krogh & Hertz, 1992), dropout (Srivastava et al., 2014), the information bottleneck (Tishby et al., 2000; Tishby & Zaslavsky, 2015), and Bayes-by-Backprop (Blundell et al., 2015) are used to balance the network complexity and the information in the data. The aim of meta-regularization is different. It governs the model complexity to avoid one complex model solving all tasks, while allowing the model's dependency on the task data to be complex. We further empirically validate this difference, finding that standard regularization techniques do not solve the memorization problem.

## 6 EXPERIMENTS

In the experimental evaluation, we aim to answer the following questions: (1) How prevalent is the memorization problem across different algorithms and domains? (2) How does the memorization problem affect the performance of algorithms on non-mutually-exclusive task distributions? (3) Is our meta-regularization approach effective for mitigating the problem and is it compatible with multiple types of meta-learning algorithms? (4) Is the problem of memorization empirically distinct from that of the standard overfitting problem?

To answer these questions, we propose several meta-learning problems involving non-mutually-exclusive task distributions, including two problems that are adapted from prior benchmarks with mutually-exclusive task distributions. We consider model-agnostic meta-learning (MAML) and conditional neural processes (CNP) as representative meta-learning algorithms. We study both variants

of our method in combination with MAML and CNP. When comparing with meta-learning algorithms with and without meta-regularization, we use the same neural network architecture, while other hyperparameters are tuned via cross-validation per-problem.

## 6.1 SINUSOID REGRESSION

First, we consider a toy sinusoid regression problem that is non-mutually-exclusive. The data for each task is created in the following way: the amplitude $A$ of the sinusoid is uniformly sampled from a set of 20 equally-spaced points $\{0.1, 0.3, \cdots, 4\}$; $u$ is sampled uniformly from $[-5, 5]$ and $y$ is sampled from $\mathcal{N}(A \sin(u), 0.1^2)$. We provide both $u$ and the amplitude $A$ (as a one-hot vector) as input, i.e. $x = (u, A)$. At the test time, we expand the range of the tasks by randomly sampling the data-generating amplitude $A$ uniformly from $[0.1, 4]$ and use a random one-hot vector for the input to the network. The meta-training tasks are a proper subset of the meta-test tasks.

Without the additional amplitude input, both MAML and CNP can easily solve the task and generalize to the meta-test tasks. However, once we add the additional amplitude input which indicates the task identity, we find that both MAML and CNP converge to the complete memorization solution and fail to generalize well to test data (Table 1 and Appendix Figures 7 and 8). Both meta-regularized MAML and CNP (MR-MAML) and (MR-CNP) instead converge to a solution that adapts to the data, and as a result, greatly outperform the unregularized methods.

Table 1: Test MSE for the non-mutually-exclusive sinusoid regression problem. We compare MAML and CNP against meta-regularized MAML (MR-MAML) and meta-regularized CNP (MR-CNP) where regularization is either on the activations (A) or the weights (W). We report the mean over 5 trials and the standard deviation in parentheses.

| Methods | MAML | MR-MAML (A) (ours) | MR-MAML (W) (ours) | CNP | MR-CNP (A) (ours) | MR-CNP (W) (ours) |
|---|---|---|---|---|---|---|
| 5 shot | 0.46 (0.04) | **0.17 (0.03)** | **0.16 (0.04)** | 0.91 (0.10) | **0.10 (0.01)** | **0.11 (0.02)** |
| 10 shot | 0.13 (0.01) | **0.07 (0.02)** | **0.06 (0.01)** | 0.92 (0.05) | **0.09 (0.01)** | **0.09 (0.01)** |

## 6.2 POSE PREDICTION

To illustrate the memorization problem on a more realistic task, we create a multi-task regression dataset based on the Pascal 3D data (Xiang et al., 2014) (See Appendix A.5.1 for a complete description). We randomly select 50 objects for meta-training and the other 15 objects for meta-testing. For each object, we use MuJoCo (Todorov et al., 2012) to render images with random orientations of the instance on a table, visualized in Figure 1. For the meta-learning algorithm, the observation $(x)$ is the $128 \times 128$ gray-scale image and the label $(y)$ is the orientation relative to a fixed canonical pose. Because the number of objects in the meta-training dataset is small, it is straightforward for a single network to memorize the canonical pose for each training object and to infer the orientation from the input image, thus achieving a low meta-training error without using $\mathcal{D}$. However, this solution performs poorly at the test time because it has not seen the novel objects and their canonical poses.

**Optimization modes and hyperparameter sensitivity.** We choose the learning rate from $\{0.0001, 0.0005, 0.001\}$ for each method, $\beta$ from $\{10^{-6}, 10^{-5}, \cdots, 1\}$ for meta-regularization and report the results with the best hyperparameters (as measured on the meta-validation set) for each method. In this domain, we find that the convergence point of the meta-learning algorithm is determined by both the optimization landscape of the objective and the training dynamics, which vary due to stochastic gradients and the random initialization. In particular, we observe that there are two modes of the objective, one that corresponds to complete memorization and one that corresponds to successful adaptation to the task data. As illustrated in the Appendix, we find that models that converge to a memorization solution have lower training error than solutions which use the task training data, indicating a clear need for meta-regularization. When the meta-regularization is on the activations, the solution that the algorithms converge to depends on the learning rate, while MR on the weights consistently converges to the adaptation solution (See Appendix Figure 9 for the sensitivity analysis). This suggests that MR on the activations is not always successful at preventing memorization. Our hypothesis is that there exists a solution in which the bottlenecked activations encode only the prediction $y^*$, and discard other information. Such a solution can achieve both low training MSE and low regularization loss without using task training data, particularly if the predicted label contains a small number of bits (i.e., because the *activations* will have low information complexity).

However, note that this solution does not achieve low regularization error when applying MR to the weights because the *function* needed to produce the predicted label does not have low information complexity. As a result, meta-regularization on the weights does not suffer from this pathology and is robust to different learning rates. Therefore, we will use regularization on weights as the proposed methodology in the following experiments and algorithms in Appendix A.1.

**Quantitative results.** We compare MAML and CNP with their meta-regularized versions (Table 2). We additionally include fine-tuning as baseline, which trains a single network on all the instances jointly, and then fine-tunes on the task training data. Meta-learning with meta-regularization (on weights) outperforms all competing methods by a large margin. We show test error as a function of the meta-regularization coefficient $\beta$ in Appendix Figure 2. The curve reflects the trade-off when changing the amount of information contained in the weights. This indicates that $\beta$ gives a knob that allows us to tune the degree to which the model uses the data to adapt versus relying on the prior.

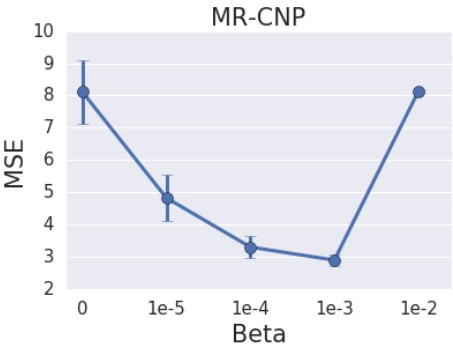 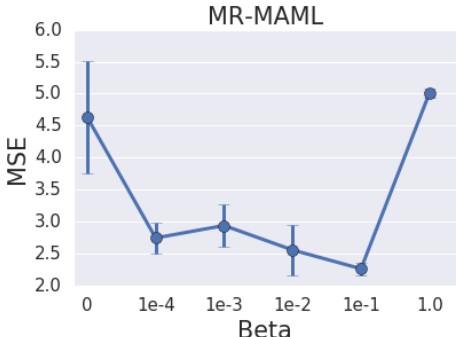

Figure 2: The performance of MAML and CNP with meta-regularization on the weights, as a function of the regularization strength $\beta$. We observe $\beta$ provides us a knob with which we can control the degree to which the algorithm adapts versus memorizes. When $\beta$ is small, we observe memorization, leading to large test error; when $\beta$ is too large, the network does not store enough information in the weights to perform the task. Crucially, in the middle of these two extremes, meta-regularization is effective in inducing adaptation, leading to good generalization. The plot shows the mean and standard deviation across 5 meta-training runs.

Table 2: Meta-test MSE for the pose prediction problem. We compare MR-MAML (ours) with conventional MAML and fine-tuning (FT). We report the average over 5 trials and standard deviation in parentheses.

| Method | MAML | MR-MAML (W) (ours) | CNP | MR-CNP (W) (ours) | FT | FT + Weight Decay |
|---|---|---|---|---|---|---|
| MSE | 5.39 (1.31) | **2.26 (0.09)** | 8.48 (0.12) | 2.89 (0.18) | 7.33 (0.35) | 6.16 (0.12) |

**Comparison to standard regularization.** We compare our meta-regularization with standard regularization techniques, weight decay (Krogh & Hertz, 1992) and Bayes-by-Backprop (Blundell et al., 2015), in Table 3. We observe that simply applying standard regularization to all the weights, as in conventional supervised learning, does not solve the memorization problem, which validates that the memorization problem differs from the standard overfitting problem.

Table 3: Meta-testing MSE for the pose prediction problem. We compare MR-CNP (ours) with conventional CNP, CNP with weight decay, and CNP with Bayes-by-Backprop (BbB) regularization on all the weights. We report the average over 5 trials and standard deviation in parentheses.

| Methods | CNP | CNP + Weight Decay | CNP + BbB | MR-CNP (W) (ours) |
|---|---|---|---|---|
| MSE | 8.48 (0.12) | 6.86 (0.27) | 7.73 (0.82) | **2.89 (0.18)** |

### 6.3 OMNIGLOT AND MINIIMAGENET CLASSIFICATION

Next, we study memorization in the few-shot classification problem by adapting the few-shot Omniglot (Lake et al., 2011) and MiniImagenet (Ravi & Larochelle, 2016; Vinyals et al., 2016) bench-

marks to the non-mutually-exclusive setting. In the *non-mutually-exclusive* N-way K-shot classification problem, each class is (randomly) assigned a fixed classification label from 1 to N. For each task, we randomly select a corresponding class for each classification label and $K$ task training data points and $K$ task test data points from that class[2]. This ensures that each class takes only one classification label across tasks and different tasks are non-mutually-exclusive (See Appendix A.5.2 for details).

We evaluate MAML, TAML (Jamal & Qi, 2019), MR-MAML (ours), fine-tuning, and a nearest neighbor baseline on non-mutually-exclusive classification tasks (Table 4). We find that MR-MAML significantly outperforms previous methods on all of these tasks. To better understand the problem, for the MAML variants, we calculate the pre-update accuracy (before adaptation on the task training data) on the meta-training data in Appendix Table 5. The high pre-update meta-training accuracy and low meta-test accuracy are evidence of the memorization problem for MAML and TAML, indicating that it is learning a model that ignores the task data. In contrast, MR-MAML successfully controls the pre-update accuracy to be near chance and encourages the learner to use the task training data to achieve low meta-training error, resulting in good performance at meta-test time.

Finally, we verify that meta-regularization does not degrade performance on the standard mutually-exclusive task. We evaluate performance as a function of regularization strength on the standard 20-way 1-shot Omniglot task (Appendix Figure 10), and we find that small values of $\beta$ lead to slight improvements over MAML. This indicates that meta-regularization substantially improves performance in the non-mutually-exclusive setting without degrading performance in other settings.

Table 4: Meta-test accuracy on non-mutually-exclusive (NME) classification. The fine-tuning and nearest-neighbor baseline results for MiniImagenet are from (Ravi & Larochelle, 2016).

| NME Omniglot | 20-way 1-shot | 20-way 5-shot |
|---|---|---|
| MAML | 7.8 (0.2)% | 50.7 (22.9)% |
| TAML (Jamal & Qi, 2019) | 9.6 (2.3)% | 67.9 (2.3)% |
| MR-MAML (W) (ours) | **83.3 (0.8)**% | **94.1 (0.1)**% |

| NME MiniImagenet | 5-way 1-shot | 5-way 5-shot |
|---|---|---|
| Fine-tuning | 28.9 (0.5))% | 49.8 (0.8))% |
| Nearest-neighbor | 41.1 (0.7)% | 51.0 (0.7) % |
| MAML | 26.3 (0.7)% | 41.6 (2.6)% |
| TAML (Jamal & Qi, 2019) | 26.1 (0.6)% | 44.2 (1.7)% |
| MR-MAML (W) (ours) | **43.6 (0.6)**% | **53.8 (0.9)**% |

## 7 CONCLUSION AND DISCUSSION

Meta-learning has achieved remarkable success in few-shot learning problems. However, we identify a pitfall of current algorithms: the need to create task distributions that are mutually exclusive. This requirement restricts the domains that meta-learning can be applied to. We formalize the failure mode, i.e. the memorization problem, that results from training on non-mutually-exclusive tasks and distinguish it as a function-level overfitting problem compared to the the standard label-level overfitting in supervised learning.

We illustrate the memorization problem with different meta-learning algorithms on a number of domains. To address the problem, we propose an algorithm-agnostic meta-regularization (MR) approach that leverages an information-theoretic perspective of the problem. The key idea is that by placing a soft restriction on the information flow from meta-parameters in prediction of test set labels, we can encourage the meta-learner to use task training data during meta-training. We achieve this by successfully controlling the complexity of model prior to the task adaptation.

The memorization issue is quite broad and is likely to occur in a wide range of real-world applications, for example, personalized speech recognition systems, learning robots that can adapt to different environments (Nagabandi et al., 2018), and learning goal-conditioned manipulation skills using trial-and-error data. Further, this challenge may also be prevalent in other conditional prediction problems, beyond meta-learning, an interesting direction for future study. By both recognizing the challenge of memorization and developing a general and lightweight approach for solving it, we believe that this work represents an important step towards making meta-learning algorithms applicable to and effective on any problem domain.

---

[2]We assume that the number of classes in the meta-training set is larger than $N$.

## ACKNOWLEDGEMENT

The authors would like to thank Alexander A. Alemi, Kevin Murphy, Luke Metz, Abhishek Kumar and the anonymous reviewers for helpful discussions and feedback. M. Yin and M. Zhou acknowledge the support of the U.S. National Science Foundation under Grant IIS-1812699.

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

# A APPENDIX

## A.1 ALGORITHM

We present the detailed algorithm for meta-regularization on weights with conditional neural processes (CNP) in Algorithm 1 and with model-agnostic meta-learning (MAML) in Algorithm 2. For CNP, we add the regularization on the weights $\theta$ of encoder and leave other weights $\tilde{\theta}$ unrestricted. For MAML, we similarly regularize the weights $\theta$ from input to an intermediate hidden layer and leave the weights $\tilde{\theta}$ for adaptation unregularized. In this way, we restrict the complexity of the pre-adaptation model not the post-adaptation model.

---

**Algorithm 1:** Meta-Regularized CNP

**input :** Task distribution $p(\mathcal{T})$; Encoder weights distribution $q(\theta; \tau) = \mathcal{N}(\theta; \tau)$ with Gaussian parameters $\tau = (\theta_\mu, \theta_\sigma)$; Prior distribution $r(\theta)$ and Lagrangian multiplier $\beta$; $\tilde{\theta}$ that parameterizes feature extractor $h_{\tilde{\theta}}(\cdot)$ and decoder $T_{\tilde{\theta}}(\cdot)$. Stepsize $\alpha$.

**output:** Network parameter $\tau, \tilde{\theta}$.

Initialize $\tau, \tilde{\theta}$ randomly;
**while** *not converged* **do**
  Sample a mini-batch of $\{\mathcal{T}_i\}$ from $p(\mathcal{T})$;
  Sample $\theta \sim q(\theta; \tau)$ with reparameterization ;
  **for** *all* $\mathcal{T}_i \in \{\mathcal{T}_i\}$ **do**
    Sample $\mathcal{D}_i = (\boldsymbol{x}_i, \boldsymbol{y}_i), \mathcal{D}_i^* = (\boldsymbol{x}_i^*, \boldsymbol{y}_i^*)$ from $\mathcal{T}_i$ ;
    Encode observation $\boldsymbol{z}_i = g_\theta(\boldsymbol{x}_i), \boldsymbol{z}_i^* = g_\theta(\boldsymbol{x}_i^*)$ ;
    Compute task context $\phi_i = a(h_{\tilde{\theta}}(\boldsymbol{z}_i, \boldsymbol{y}_i))$ with aggregator $a(\cdot)$;

  Update $\tilde{\theta} \leftarrow \tilde{\theta} + \alpha \nabla_{\tilde{\theta}} \sum_{\mathcal{T}_i} \log q(\boldsymbol{y}_i^* | T_{\tilde{\theta}}(\boldsymbol{z}_i^*, \phi_i))$ ;
  Update $\tau \leftarrow \tau + \alpha \nabla_\tau [\sum_{\mathcal{T}_i} \log q(\boldsymbol{y}_i^* | T_{\tilde{\theta}}(\boldsymbol{z}_i^*, \phi_i)) - \beta D_{\text{KL}}(q(\theta; \tau) || r(\theta))]$

---

**Algorithm 2:** Meta-Regularized MAML

**input :** Task distribution $p(\mathcal{T})$; Weights distribution $q(\theta; \tau) = \mathcal{N}(\theta; \tau)$ with Gaussian parameters $\tau = (\theta_\mu, \theta_\sigma)$; Prior distribution $r(\theta)$ and Lagrangian multiplier $\beta$; Stepsize $\alpha, \alpha'$.

**output:** Network parameter $\tau, \tilde{\theta}$.

Initialize $\tau, \tilde{\theta}$ randomly;
**while** *not converged* **do**
  Sample a mini-batch of $\{\mathcal{T}_i\}$ from $p(\mathcal{T})$;
  Sample $\theta \sim q(\theta; \tau)$ with reparameterization ;
  **for** *all* $\mathcal{T}_i \in \{\mathcal{T}_i\}$ **do**
    Sample $\mathcal{D}_i = (\boldsymbol{x}_i, \boldsymbol{y}_i), \mathcal{D}_i^* = (\boldsymbol{x}_i^*, \boldsymbol{y}_i^*)$ from $\mathcal{T}_i$ ;
    Encode observation $\boldsymbol{z}_i = g_\theta(\boldsymbol{x}_i), \boldsymbol{z}_i^* = g_\theta(\boldsymbol{x}_i^*)$ ;
    Compute task specific parameter $\phi_i = \tilde{\theta} + \alpha' \nabla_{\tilde{\theta}} \log q(\boldsymbol{y}_i | \boldsymbol{z}_i, \tilde{\theta})$ ;

  Update $\tilde{\theta} \leftarrow \tilde{\theta} + \alpha \nabla_{\tilde{\theta}} \sum_{\mathcal{T}_i} \log q(\boldsymbol{y}_i^* | \boldsymbol{z}_i^*, \phi_i)$ ;
  Update $\tau \leftarrow \tau + \alpha \nabla_\tau [\sum_{\mathcal{T}_i} \log q(\boldsymbol{y}_i^* | \boldsymbol{z}_i^*, \phi_i) - \beta D_{\text{KL}}(q(\theta; \tau) || r(\theta))]$

---

---

**Algorithm 3:** Meta-Regularized Methods in Meta-testing

---

**input** : Meta-testing task $\mathcal{T}$ with training data $\mathcal{D} = (\boldsymbol{x}, \boldsymbol{y})$ and testing input $\boldsymbol{x}^*$, optimized parameters $\tau, \tilde{\theta}$.

**output:** Prediction $\hat{y}^*$

---

**for** $k$ *from 1 to K* **do**

    Sample $\theta_k \sim q(\theta; \tau)$;

    Encode observation $\boldsymbol{z}_k = g_{\theta_k}(\boldsymbol{x})$, $\boldsymbol{z}_k^* = g_{\theta_k}(\boldsymbol{x}^*)$ ;

    Compute task specific parameter $\phi_k = a(h_{\tilde{\theta}}(\boldsymbol{z}_k, \boldsymbol{y}))$ for MR-CNP and

    $\phi_k = \tilde{\theta} + \alpha' \nabla_{\tilde{\theta}} \log q(\boldsymbol{y}|\boldsymbol{z}_k, \tilde{\theta})$ for MR-MAML;

    Predict $\hat{y}_k^* \sim q(\hat{y}^*|z_k^*, \phi_k, \tilde{\theta})$

Return prediction $\hat{y}^* = \frac{1}{K} \sum_{k=1}^{K} \hat{y}_k^*$

---

### A.2 META REGULARIZATION ON ACTIVATIONS

We show that $I(x^*; \hat{y}^*|z^*, \theta) \leq I(\hat{y}^*; \mathcal{D}|z^*, \theta)$. By Figure 4, we have that $I(\hat{y}^*; x^*|\theta, \mathcal{D}, z^*) = 0$. By the chain rule of mutual information we have

$$
\begin{aligned}
I(\hat{y}^*; \mathcal{D}|z^*, \theta) &= I(\hat{y}^*; \mathcal{D}|z^*, \theta) + I(\hat{y}^*; x^*|\mathcal{D}, \theta, z^*) \\
&= I(\hat{y}^*; x^*, \mathcal{D}|\theta, z^*) \\
&= I(x^*; \hat{y}^*|\theta, z^*) + I(\hat{y}^*; \mathcal{D}|x^*, \theta, z^*) \\
&\geq I(x^*; \hat{y}^*|\theta, z^*)
\end{aligned}
\tag{10}
$$

### A.3 META REGULARIZATION ON WEIGHTS

Similar to (Achille & Soatto, 2018), we use $\xi$ to denote the unknown parameters of the true data generating distribution. This defines a joint distribution $p(\xi, \mathcal{M}, \theta) = p(\xi)p(\mathcal{M}|\xi)q(\theta|\mathcal{M})$. Furthermore, we have a predictive distribution $q(\hat{y}^*|x^*, \mathcal{D}, \theta) = \mathbb{E}_{\phi|\theta, \mathcal{D}} [q(\hat{y}^*|x^*, \phi, \theta)]$.

The meta-training loss in Eq. 1 is an upper bound for the cross entropy $H_{p,q}(y_{1:N}^*|x_{1:N}^*, \mathcal{D}_{1:N}, \theta)$. Using an information decomposition of cross entropy (Achille & Soatto, 2018), we have

$$
\begin{aligned}
H_{p,q}(y_{1:N}^*|x_{1:N}^*, \mathcal{D}_{1:N}, \theta) = {} & H(y_{1:N}^*|x_{1:N}^*, \mathcal{D}_{1:N}, \xi) + I(\xi; y_{1:N}^*|x_{1:N}^*, \mathcal{D}_{1:N}, \theta) \\
& + \mathbb{E}\left[D_{\mathrm{KL}}(p(y_{1:N}^*|x_{1:N}^*, \mathcal{D}_{1:N}, \theta)||q(y_{1:N}^*|x_{1:N}^*, \mathcal{D}_{1:N}, \theta))\right] + I(\mathcal{D}_{1:N}; \theta|x_{1:N}^*, \xi) \\
& - I(y_{1:N}^*, \mathcal{D}_{1:N}; \theta|x_{1:N}^*, \xi).
\end{aligned}
\tag{11}
$$

Here the only negative term is the $I(y_{1:N}^*, \mathcal{D}_{1:N}; \theta|x_{1:N}^*, \xi)$, which quantifies the information that the meta-parameters contain about the meta-training data beyond what can be inferred from the data generating parameters (i.e., memorization). Without proper regularization, the cross entropy loss can be minimized by maximizing this term. We can control its value by upper bounding it

$$
\begin{aligned}
I(y_{1:N}^*, \mathcal{D}_{1:N}; \theta|x_{1:N}^*, \xi) &= \mathbb{E}\left[\log \frac{q(\theta|\mathcal{M}, \xi)}{q(\theta|x_{1:N}^*, \xi)}\right] \\
&= \mathbb{E}\left[\log \frac{q(\theta|\mathcal{M})}{q(\theta|x_{1:N}^*, \xi)}\right] \\
&= \mathbb{E}\left[D_{\mathrm{KL}}(q(\theta|\mathcal{M})||q(\theta|x_{1:N}^*, \xi))\right] \\
&\leq \mathbb{E}\left[D_{\mathrm{KL}}(q(\theta|\mathcal{M})||r(\theta))\right],
\end{aligned}
$$

where the second equality follows because $\theta$ and $\xi$ are conditionally independent given $\mathcal{M}$. This gives the regularization in Section 4.2.

### A.4 PROOF OF THE PAC-BAYES GENERALIZATION BOUND

First, we prove a more general result and then specialize it. The goal of the meta-learner is to extract information about the meta-training tasks and the test task training data to serve as a prior for test examples from the novel task. This information will be in terms of a distribution $Q$ over possible models. When learning a new task, the meta-learner uses the training task data $\mathcal{D}$ and a model

parameterized by $\theta$ (sampled from $Q(\theta)$) and outputs a distribution $q(\phi|\mathcal{D}, \theta)$ over models. Our goal is to learn $Q$ such that it performs well on novel tasks.

To formalize this, define

$$er(Q, \mathcal{D}, \mathcal{T}) = \mathbb{E}_{\theta \sim Q(\theta), \phi \sim q(\phi|\theta, \mathcal{D}), (x^*, y^*) \sim p(x, y|\mathcal{T})} \left[ \mathcal{L}(\phi(x^*), y^*) \right] \tag{12}$$

where $\mathcal{L}(\phi(x^*), y^*)$ is a bounded loss in $[0, 1]$. Then, we would like to minimize the error on novel tasks

$$er(Q) = \min_Q \mathbb{E}_{\mathcal{T} \sim p(\mathcal{T}), \mathcal{D} \sim p(x, y|\mathcal{T})} \left[ er(Q, \mathcal{D}, \mathcal{T}) \right] \tag{13}$$

Because we only have a finite training set, computing $er(Q)$ is intractable, but we can form an empirical estimate:

$$\hat{er}(Q, \mathcal{D}_1, \mathcal{D}_1^*, ..., \mathcal{D}_n, \mathcal{D}_n^*) = \frac{1}{n} \sum_{i=1}^{n} \underbrace{\mathbb{E}_{\theta \sim Q(\theta), \phi_i \sim q(\phi|\theta, \mathcal{D}_i)} \left[ \frac{1}{K} \sum_{(x^*, y^*) \in \mathcal{D}_i^*} \mathcal{L}(\phi(x^*), y^*) \right]}_{\hat{er}(Q, \mathcal{D}_i, \mathcal{D}_i^*)} \tag{14}$$

where for exposition we assume $K = |\mathcal{D}_i^*|$ is the same for all $i$. We would like to relate $er(Q)$ and $\hat{er}(Q, \mathcal{D}_1, \mathcal{D}_1^*, ..., \mathcal{D}_n, \mathcal{D}_n^*)$, but the challenge is that $Q$ may depend on $\mathcal{D}_1, \mathcal{D}_1^*, ..., \mathcal{D}_n, \mathcal{D}_n^*$ due to the learning algorithm. There are two sources of generalization error: (i) error due to the finite number of observed tasks and (ii) error due to the finite number of examples observed per task. Closely following the arguments in (Amit & Meir, 2018), we apply a standard PAC-Bayes bound to each of these and combine the results with a union bound.

**Theorem.** *Let $Q(\theta)$ be a distribution over parameters $\theta$ and let $P(\theta)$ be a prior distribution. Then for any $\delta \in (0, 1]$, with probability at least $1 - \delta$, the following inequality holds uniformly for all distributions $Q$,*

$$er(Q) \leq \frac{1}{n} \sum_{i=1}^{n} \hat{er}(Q, \mathcal{D}_i, \mathcal{D}_i^*) + \left( \sqrt{\frac{1}{2(K-1)}} + \sqrt{\frac{1}{2(n-1)}} \right) \sqrt{D_{KL}(Q\|P) + \log \frac{n(K+1)}{\delta}} \tag{15}$$

*Proof.* To start, we state a classical PAC-Bayes bound and use it to derive generalization bounds on task and datapoint level generalization, respectively.

**Theorem 2.** *Let $\mathcal{X}$ be a sample space (i.e. a space of possible datapoints). Let $P(X)$ be a distribution over $\mathcal{X}$ (i.e. a data distribution). Let $\Theta$ be a hypothesis space. Given a "loss function" $l(\theta, X) : \Theta \times \mathcal{X} \to [0, 1]$ and a collection of $M$ i.i.d. random variables sampled from $P(X)$, $X_1, ..., X_M$, let $\pi$ be a prior distribution over hypotheses in $\Theta$ that does not depend on the samples but may depend on the data distribution $P(X)$. Then, for any $\delta \in (0, 1]$, the following bound holds uniformly for all posterior distributions $\rho$ over $\Theta$*

$$P\Big( \mathbb{E}_{X_i \sim P(X), \theta \sim \rho(\cdot)} [l(\theta, X_i)] \leq \frac{1}{M} \sum_{m=1}^{M} \mathbb{E}_{\theta \sim \rho(\cdot)} [l(\theta, X_m)] + \sqrt{\frac{1}{2(M-1)} \left( D_{KL}(\rho\|\pi) + \log \frac{M}{\delta} \right)}, \forall \rho \Big)$$
$$\geq 1 - \delta. \tag{16}$$

**Meta-level generalization** First, we bound the task-level generalization, that is we relate $er(Q)$ to $\frac{1}{n} \sum_{i=1}^{n} er(Q, \mathcal{D}_i, \mathcal{T}_i)$. Letting the samples be $X_i = (\mathcal{D}_i, \mathcal{T}_i)$, and $l(\theta, X_n) = \mathbb{E}_{\phi_i \sim q(\phi|\mathcal{D}_i, \theta), (x^*, y^*) \sim \mathcal{T}_i} [\mathcal{L}(\phi(x^*), y^*)]$, then Theorem 1 says that for any $\delta_0 \sim (0, 1]$

$$P\left( er(Q) \leq \frac{1}{n} \sum_{i=1}^{n} er(Q, \mathcal{D}_i, \mathcal{T}_i) + \sqrt{\frac{1}{2(n-1)} \left( D_{KL}(Q\|P) + \log \frac{n}{\delta_0} \right)}, \forall Q \right) \geq 1 - \delta_0, \tag{17}$$

where $P$ is a prior over $\theta$.

**Within task generalization** Next, we relate $er(Q, \mathcal{D}_i, \mathcal{T}_i)$ to $\hat{er}(Q, \mathcal{D}_i, \mathcal{D}_i^*)$ via the PAC-Bayes bound. For a fixed task $i$, task training data $\mathcal{D}_i$, a prior $\pi(\phi|\mathcal{T}_i)$ that only depends on the training

data, and any $\delta_i \in (0, 1]$, we have that

$$
P\Big(\mathbb{E}_{(x^*,y^*)\sim p(x,y|\mathcal{T}_i)\rho(\phi_i)}\left[\mathcal{L}(\phi_i(x^*),y^*)\right] \leq \mathbb{E}_{\rho(\phi_i)}\left[\frac{1}{K}\sum_{(x^*,y^*)\in\mathcal{D}_i^*}\mathcal{L}(\phi_i(x^*),y^*)\right]
$$
$$
+ \sqrt{\frac{1}{2(K-1)}\left(D_{KL}(\rho||\pi) + \log\frac{K}{\delta_i}\right)}, \forall\rho\Big) \geq 1 - \delta_i.
$$

Now, we choose $\pi(\phi|\mathcal{T}_i)$ to be $\int P(\theta)q(\phi|\theta,\mathcal{D}_i)d\theta$ and restrict $\rho(\phi)$ to be of the form $\int Q(\theta)q(\phi|\theta,\mathcal{D}_i)d\theta$ for any $Q$. While, $\pi$ and $\rho$ may be complicated distributions (especially, if they are defined implicitly), we know that with this choice of $\pi$ and $\rho$, $D_{KL}(\rho||\pi) \leq D_{KL}(Q||P)$ (Cover & Thomas, 2012), hence, we have

$$
P\left(er(Q,\mathcal{D}_i,\mathcal{T}_i) \leq \hat{er}(Q,\mathcal{D}_i,\mathcal{D}_i^*) + \sqrt{\frac{1}{2(K-1)}\left(D_{KL}(Q||P) + \log\frac{K}{\delta_i}\right)}, \forall Q\right) \geq 1 - \delta_i
$$

(18)

**Overall bound on meta-learner generalization** Combining Eq. (17) and (18) using the union bound, we have

$$
P\Big(er(Q) \leq \frac{1}{n}\sum_{i=1}^{n}\hat{er}(Q,\mathcal{D}_i,\mathcal{D}_i^*) + \sqrt{\frac{1}{2(K-1)}D_{KL}(Q||P) + \log\frac{K}{\delta_i}}
$$
$$
+ \sqrt{\frac{1}{2(n-1)}D_{KL}(Q||P) + \log\frac{n}{\delta_0}}, \forall Q\Big) \geq 1 - \left(\sum_i \delta_i + \delta_0\right)
$$

(19)

Choosing $\delta_0 = \frac{\delta}{K+1}$ and $\delta_i = \frac{K\delta}{n(K+1)}$, then we have:

$$
P\Big(er(Q) \leq \frac{1}{n}\sum_{i=1}^{n}\hat{er}(Q,\mathcal{D}_i,\mathcal{D}_i^*) + \left(\sqrt{\frac{1}{2(K-1)}} + \sqrt{\frac{1}{2(n-1)}}\right)\sqrt{D_{KL}(Q||P) + \log\frac{n(K+1)}{\delta}}, \forall Q\Big)
$$
$$
\geq 1 - \delta.
$$

(20)

$\square$

Because $n$ is generally large, by Taylor expansion of the complexity term we have

$$
\left(\sqrt{\frac{1}{2(K-1)}} + \sqrt{\frac{1}{2(n-1)}}\right)\sqrt{\left(D_{KL}Q||P) + \log\frac{n(K+1)}{\delta}\right)}
$$
$$
= \frac{1}{2\sqrt{\log n(K+1)/\delta}}\left(\sqrt{\frac{1}{2(K-1)}} + \sqrt{\frac{1}{2(n-1)}}\right)\left(D_{KL}Q||P) + 2\log(\frac{n(K+1)}{\delta})\right) + o(1)
$$

Re-defining the coefficient of KL term as $\beta$ and omitting the constant and higher order term, we recover the meta-regularization bound in Eq.(5) when $Q(\theta) = \mathcal{N}(\theta; \theta_\mu, \theta_\sigma)$.

## A.5 EXPERIMENTAL DETAILS

### A.5.1 POSE PREDICTION

We create a multi-task regression dataset based on the Pascal 3D data (Xiang et al., 2014). The dataset consists of 10 classes of 3D object such as "aeroplane", "sofa", "TV monitor", etc. Each class has multiple different objects and there are 65 objects in total. We randomly select 50 objects for meta-training and the other 15 objects for meta-testing. For each object, we use MuJoCo (Todorov et al., 2012) to render 100 images with random orientations of the instance on a table, visualized in Figure 1. For the meta-learning algorithm, the observation $(x)$ is the $128 \times 128$ gray-scale image and the label $(y)$ is the orientation re-scaled to be within $[0, 10]$. For each task, we randomly sample

30 $(x, y)$ pairs for an object and evenly split them between task training and task test data. We use a meta batch-size of 10 tasks per iteration.

For MR-CNP, we use a convolutional encoder with a fully connected bottom layer to map the input image to a 20-dimensional latent representation $z$ and $z^*$ for task training input $x$ and test input $x^*$ respectively. The $(z, y)$ are concatenated and mapped by the feature extractor and aggregator which are fully connected networks to the 200 dimensional task summary statistics $\phi$. The decoder is a fully connected network that maps $(\phi, z^*)$ to the prediction $\hat{y}^*$.

For MR-MAML, we use a convolutional encoder to map the input image to a $14 \times 14$ dimensional latent representation $z$ and $z^*$. The pairs $(z, y)$ are used in the task adaptation step to get a task specific parameter $\phi$ via gradient descent. Then $z^*$ is mapped to the prediction $\hat{y}^*$ with a convolutional predictor parameterized by $\phi$. The network is trained using 5 gradient steps with learning rate 0.01 in the inner loop for adaptation and evaluated using 20 gradient steps at the test-time.

### A.5.2 NON-MUTUALLY-EXCLUSIVE CLASSIFICATION

The Omniglot dataset consists of 20 instances of 1623 characters from 50 different alphabets. We randomly choose 1200 characters for meta-training and use the remaining for testing. The meta-training characters are partitioned into 60 disjoint sets for 20-way classification. The MiniImagenet dataset contains 100 classes of images including 64 training classes, 12 validation classes, and 24 test classes. We randomly partition the 64 meta-training classes into 13 disjoint sets for 5-way classification with one label having one less class of images than the others.

For MR-MAML we use a convolutional encoder similar to the pose prediction problem. The dimension of $z$ and $z^*$ is $14 \times 14$ for Omniglot and $20 \times 20$ for MiniImagenet. We use a convolutional decoder for both datasets. Following (Finn et al., 2017), we use a meta batch-size of 16 for 20-way Omniglot classification and meta batch-size of 4 for 5-way MiniImagenet classification. The meta-learning rate is chosen from $\{0.001, 0.005\}$ and the $\beta$ for meta-regularized methods are chosen from $\{10^{-7}, 10^{-6}, \dots, 10^{-3}\}$. The optimal hyperparameters are chosen for each method separately via cross-validation.

### A.6 ADDITIONAL ILLUSTRATION AND GRAPHICAL MODEL

We show a standard few-shot classification setup in meta-learning to illustrate a mutually-exclusive task distribution and a graphical model for the regularization on the activations.

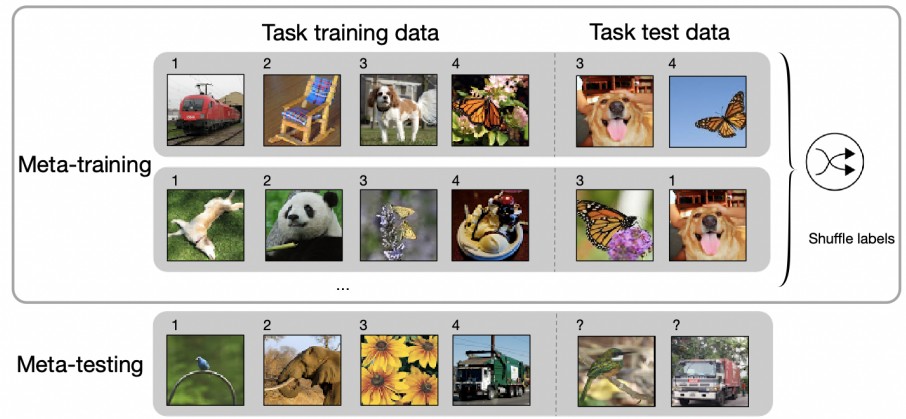

Figure 3: An example of *mutually-exclusive* task distributions. In each task of mutually-exclusive few-shot classification, different classes are randomly assigned to the $N$-way classification labels. The same class, such as the dog and butterfly in this illustration, can be assigned different labels across tasks which makes it impossible for one model to solve all tasks simultaneously.

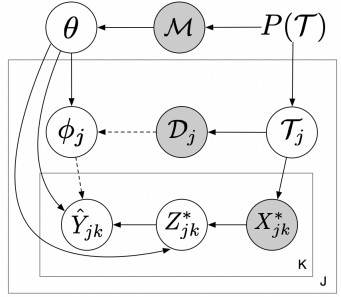

Figure 4: Graphical model of the regularization on activations. Observed variables are shaded and $Z$ is bottleneck variable. The complete memorization corresponds to the graph without the dashed arrows.

## A.7    ADDITIONAL RESULTS

As shown in Figures 5, 7 and 8, when meta-learning algorithms converge to the memorization solution, the test tasks must be similar to the train tasks in order to achieve low test error. For CNP, although the task training set contains sufficient information to infer the correct amplitude, this information is ignored and the regression curve at test-time is determined by the one-hot vector. As a result, CNP can only generalize to points from the curves it has seen in the training (Figure 7 first row). On the other hand, MAML does use the task training data (Figure 5, 8 and Table 1), however, its performance is much worse than in the mutually-exclusive task. MR-MAML and MR-CNP avoid converging to a memorization solution and achieve excellent test performance on sinusoid task.

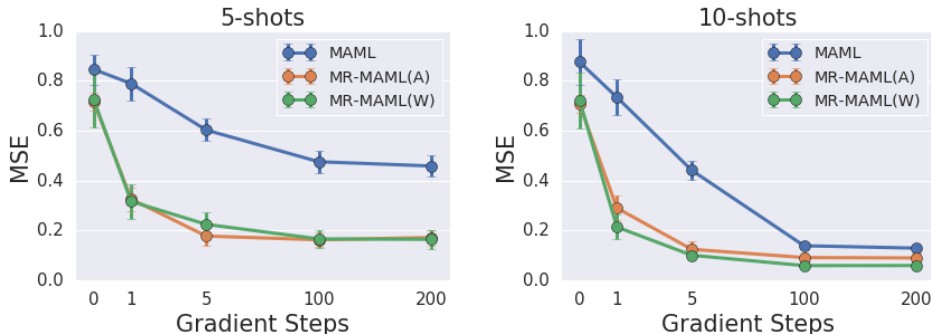

Figure 5: Test MSE on the mutually-non-exclusive sinusoid problem as function of the number of gradient steps used in the inner loop of MAML and MR-MAML. For each trial, we calculate the mean MSE over 100 randomly generated meta-testing tasks. We report the mean and standard deviation over 5 random trials.

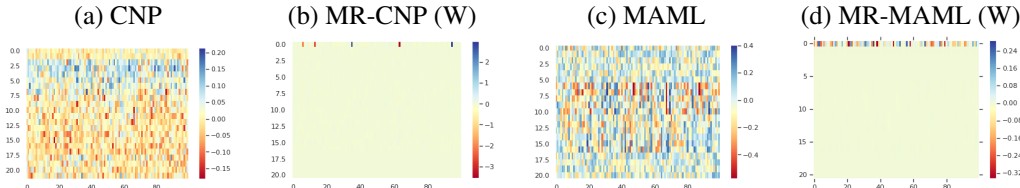

Figure 6:    Visualization of the optimized weight matrix $W$ that is connected to the inputs in the sinusoid regression example. The input $x = (u, A)$ where $u \sim \text{Unif}(-5, 5)$, $A$ is 20 dimensional one-hot vector and the intermediate layer is 100 dimensional, hence $x \in \mathbb{R}^{21}$ and $W \in \mathbb{R}^{21 \times 100}$. For both CNP and MAML, the meta-regularization restricts the part of weights that is connected to $A$ close to 0. Therefore it avoids storing the amplitude information in weights and forces the amplitude to be inferred from the task training data $\mathcal{D}$, hence preventing the memorization problem.

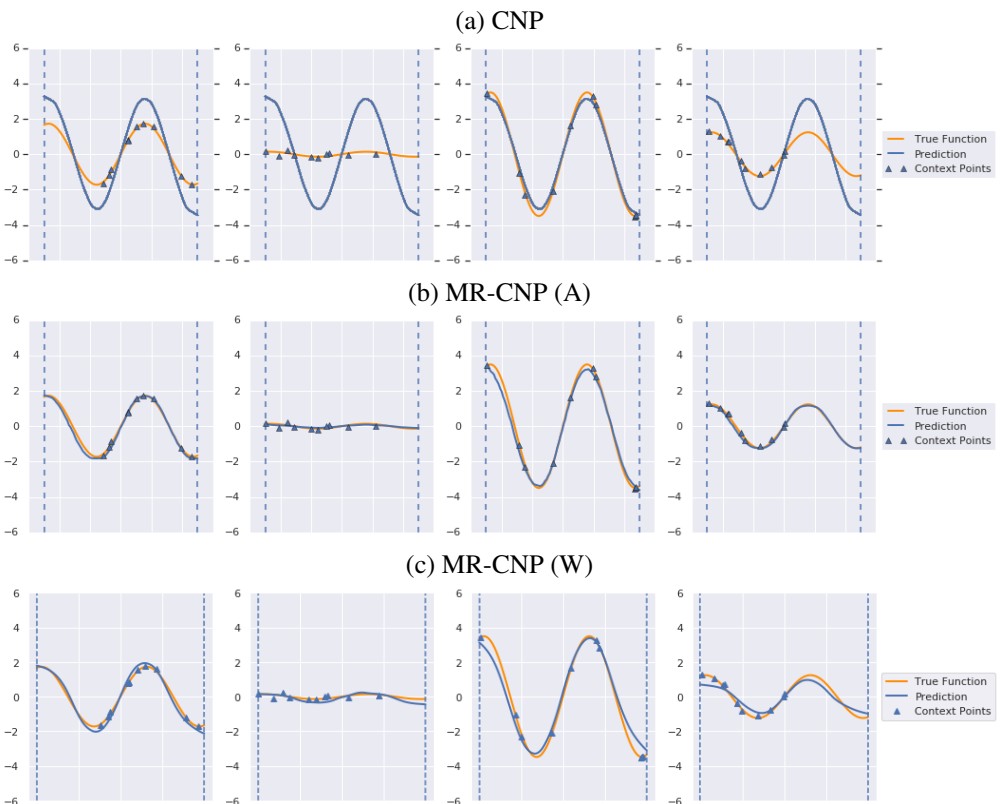

Figure 7: Meta-test results on the non-mutually-exclusive sinusoid regression problem with CNP. For each row, the amplitudes of the true curves (orange) are randomly sampled uniformly from $[0.1, 4]$. For illustrative purposes, we fix the one-hot vector component of the input. (a): The vanilla CNP cannot adapt to new task training data at test-time and the shape of prediction curve (blue) is determined by the one-hot amplitude not the task training data. (b) (c): Adding meta-regularization on both activation and weights enables the CNP to use the task training data at meta-training and causes the model to generalize well at test-time.

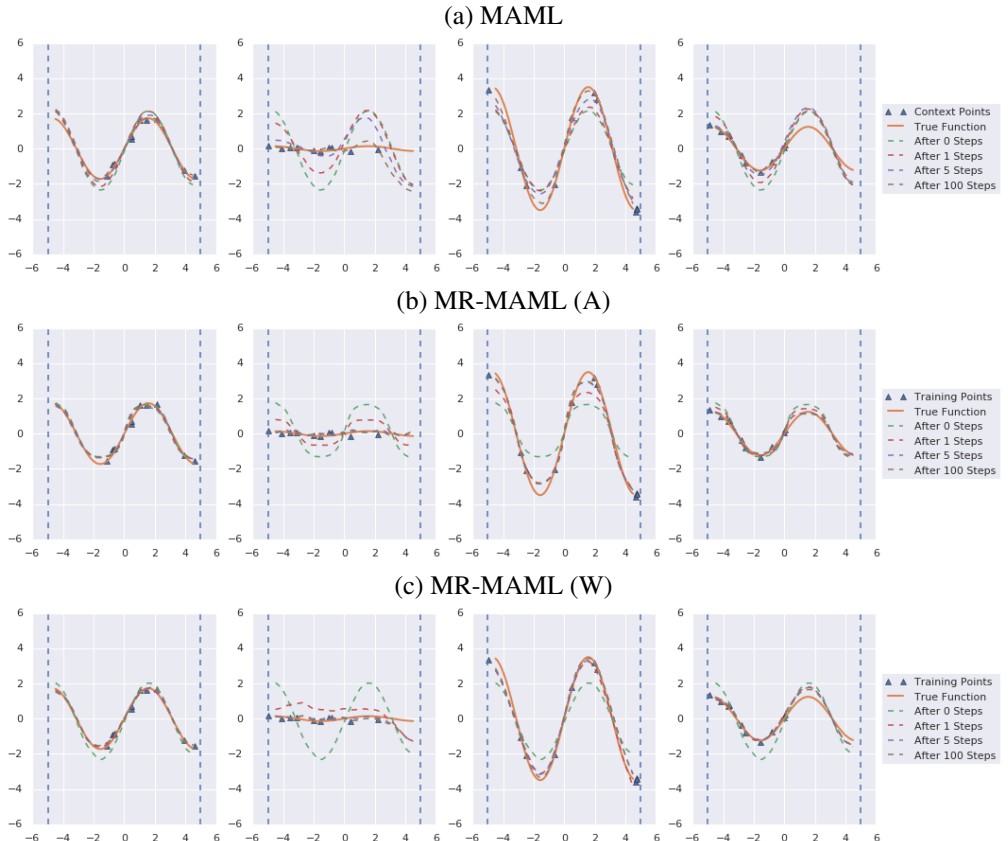

Figure 8: Meta-test results on the non-mutually-exclusive sinusoid regression problem with MAML. For each row, the true amplitudes of the true curves (orange) are randomly sampled uniformly from $[0.1, 4]$. For illustrative purposes, we fix the one-hot vector component of the input. (a): Due to memorization, MAML adapts slowly and has large generalization error at test-time. (b) (c): Adding meta-regularization on both activation and weights recovers efficient adaptation.

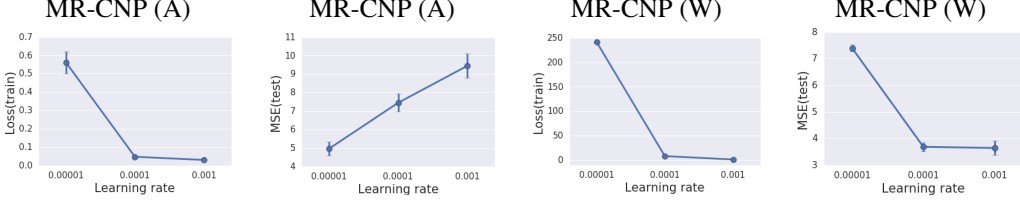

Figure 9: Sensitivity of activation regularization and weight regularization with respect to the learning rate on the pose prediction problem. For activation regularization, lower training loss corresponds to higher test MSE which indicates that the memorization solution is not solved. For weights regularization, lower training loss corresponds to lower test MSE which indicates proper training can converge to the adaptation solution.

In Table 5, we report the pre-update accuracy for the non-mutually-exclusive classification experiment in Section 6.3. The pre-update accuracy is obtained by the initial parameters $\theta$ rather than the task adapted parameters $\phi$. At the meta-training time, for both MAML and MR-MAML the post-update accuracy obtained by using $\phi$ gets close to 1. High pre-update accuracy reflects the memorization problem. For example, in 20-way 1-shot Omniglot example, the pre-update accuracy for MAML is 99.2% at the training time, which means only 0.8% improvement in accuracy is due to adaptation, so the task training data is ignored to a large extent. The pre-update training accuracy for MR-MAML is 5%, which means 95% improvement in accuracy during training is due to the adaptation. This explains why in Table 4, the test accuracy of MR-MAML is much higher than that of MAML at the test-time, since the task training data is used to achieve fast adaptation.

Table 5: Meta-training *pre-update* accuracy on non-mutually-exclusive classification. MR-MAML controls the meta-training pre-update accuracy close to random guess and achieves low training error after adaptation.

| NME Omniglot | 20-way 1-shot | 20-way 5-shot | | NME MiniImagenet | 5-way 1-shot | 5-way 5-shot |
|---|---|---|---|---|---|---|
| MAML | 99.2 (0.2)% | 45.1 (38.9)% | | MAML | 99.4 (0.1)% | 21.0(1.2)% |
| TAML | 68.9(43.1)% | 6.7 (1.8)% | | TAML | 99.4 (0.1)% | 20.8(0.4)% |
| MR-MAML (ours) | **5.0 (0)**% | **5.0 (0)**% | | MR-MAML (ours) | **20.0(0)**% | **20.2(0.1)**% |

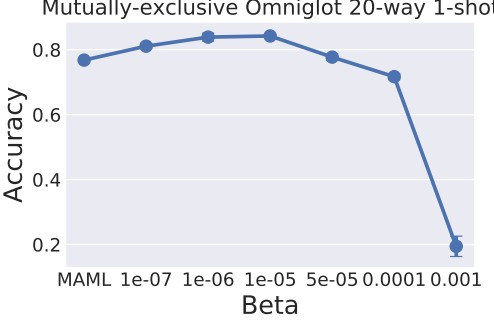

Figure 10: The test accuracy of MAML with meta-regularization on the weights as a function of the regularization strength $\beta$ on the mutually-exclusive 20-way 1-shot Omniglot problem. The plot shows the mean and standard deviation across 5 meta-training runs. When $\beta$ is small, MR-MAML slightly outperforms MAML, indicating that meta-regularization does not degrade performance on mutually-exclusive tasks. The accuracy numbers are not directly comparable to previous work (e.g., (Finn et al., 2017)) because we do not use data augmentation.

