# OpenReview forum: "Meta-Learning without Memorization"
_ICLR.cc/2020/Conference — Accept (Spotlight)_

### Official Review · AnonReviewer1 · 2019-10-22
**Official Blind Review #1**

**Rating:** 8

**Review:**

################################################################################
Summary:

This paper illustrates, identifies, and formally defines a memorization problem in meta-learning -- the model can simply memorize meta-training tasks and ignore meta-training train sets. The paper proposes to optimize the mutual information between testing predictions and the training data (given input and meta model), and upper bound it by imposing a information bottleneck between output and input+model. Unlike related work, this paper specifically is able to generalize to meta-test even when the meta-train dataset is not made confusing enough (i.e. even when model can learn well from test data in meta-train alone), making it applicable to use cases where it is hard to make the dataset confusing.

################################################################################
Decision

I vote for accepting this paper, since as far as I know this paper gives a novel insight to the overfitting problem in meta learning, and has formulated the problem formally with theoretical insight, and given a working solution with strong experiment results and clean comparative studies. Despite somewhat narrow experiments and sometimes confusing writing, the paper should provide new insight to meta-learning.

################################################################################
Pros:
! DISCLAIMER: I am not an expert in this field, so take my novelty judgements with a grain of salt.
+ Novel view into meta-learning's overfitting problem
(1) Large models can simply memorize which input data corresponds to which task, and memorize the meta-training tasks, without being able to generalize into meta-test tasks.
(2) Formulates this into a low mutual information between meta-train training data and predictions.
+ Easy to implement and quite widely applicable as a regularization loss addon to multiple existing meta-learning methods
+ Impact-wise, the paper takes meta-learning further from memorization, making methods more capable of operating on less-carefully designed, more natural datasets (rather than permutation of datasets)
+ Experiments are clean with ablation studies and hyperparameter sensitivity tests, and method performs well across real and toy datasets
+ Motivation part is easy to read

################################################################################
Cons:
- I'm still skeptical of the novelty of the paper, since the conclusions are, in hindsight, very straightforward.
- Sometimes sentences are very confusing to readers.
(1) The term "mutually-exclusive" is confusing because the view-point example the paper gives seems to be mutually exclusive (each task has its own kind of data, hence "exclusive"). It is unclear whether the task data is exclusive, or the task function is exclusive, and not straightforward to see its relationship with memorization. Consider renaming it to e.g. "mutually-confusing" or "mutually-contradictory mapping".
(2) Can you please rename "information-theoretic meta-regularizer" to "meta-regularizer using information theory"? It is hard to read for non-native speakers.
(3) Paragraph under definition 1 is confusing and has redundancies.
(4) Section 4.1, not very clear how the logic goes from the decomposition to adding the upper bound to the loss, and how the other term comes in.
- For the motivation, it is better to give examples of use cases when it is impossible to make meta-train "mutually-exclusive". I'm sure even in the patient example you can shuffle classes or input dimensions.
- An experiment comparing to others in mutually-exclusive datasets would be nice to have, in order to judge how much this compensates a badly-designed meta-learning dataset.

################################################################################
Improvements:
- Clarify each point in the "Cons" section.
- Please also clarify if all methods in all experiments are hyperparameter-tuned separately, i.e. that the experiments are not favoring the MR-* model in any way. (the paper only clarifies it in one of the experiments)
- For future work, does recent developments in mutual information modeling (e.g. MINE https://arxiv.org/abs/1801.04062) help this method in any way? e.g. try increasing mutual information between some representation of the meta-train training data and some feature vectors before the prediction?
- First parenthesis in Section 4.1 has a misplaced space



################################################################################
Post rebuttal
################################################################################
It seems that reviewers agree that the contributions are novel (regardless of whether each reviewer thinks it is trivial). So that addresses my main concern. I think the contributions are novel enough since it gives theoretical guidance as well. Other concerns are mostly addressed by the rebuttal. I will keep my rating.

Although I do urge the authors to reconsider the name choice "mutual-exclusive tasks" since it is not very informative and quite confusing to readers.


**Experience Assessment:**

I do not know much about this area.

**Review Assessment: Checking Correctness Of Derivations And Theory:**

I assessed the sensibility of the derivations and theory.

**Review Assessment: Checking Correctness Of Experiments:**

I carefully checked the experiments.

**Review Assessment: Thoroughness In Paper Reading:**

I read the paper at least twice and used my best judgement in assessing the paper.

---

> ### Author Response · Authors · 2019-11-10
> **Response to Reviewer #1, part1**
>
> Thank you for your insightful and constructive comments and suggestions. Please see our point-by-point response to your comments below.
>
> Q1)  I'm still skeptical of the novelty of the paper.
>
> A1: Our primary contributions and novelties are:
>
> i) We are the first to identify and formalize the memorization problem in meta-learning, a previously unappreciated issue. We find that its main cause is the non-mutually-exclusive task distribution.  Furthermore, as the reviewer notes, we demonstrate that it exists in multiple meta-learning algorithms and can significantly deteriorate performance. Hence, we believe the identification and formalization of this problem to be a significant contribution.
>
> ii) We propose an effective and principled regularization approach, and it is not a trivial application of existing ideas. Firstly, as revealed in Table 2 and 3, vanilla regularization on all the model parameters does not solve the memorization problem. Knowing what parameters to be regularized is important for the meta-regularizer to be effective and the theory matters. Secondly, we identify that regularization on activations can fail to prevent the memorization problem and hypothesize why. This indicates that seemingly reasonable alternatives are insufficient. Finally, we consider the simplicity of our approach an advantage because it is then compatible with multiple meta-learning algorithms and easy to implement in practice.
>
> iii) We designed and constructed a novel non-mutually-exclusive pose regression dataset which can serve as a benchmark for future algorithms.
>
> In sum, the problem we identified and studied, the methods we proposed and the pose dataset we created are all novel. We believe this paper can bring awareness of the memorization problem when developing new meta-learning methodologies or applications. Moreover, the datasets and experiments we developed can provide a benchmark for further study of the memorization problem in meta-learning.
>
>
> Q2) Some sentences are confusing.
>
> A2: (1) We call it mutually exclusive because each task has its own kind of function, i.e. the task functions are exclusive so that a single neural net cannot solve all tasks.
>
> (2) We have changed the “information-theoretic meta-regularizer” to “meta-regularizer using information theory”.
>
> (3) We use the paragraph to clarify the memorization and memorization problem. We have modified it to make it clear that if the meta-testing tasks are similar to meta-training ones, with memorization the model can generalize to new datapoints, which differs from common overfitting on datapoints. Memorization is undesirable when the problem requires using the new training data to solve the meta-test tasks. Does this clarify the confusion?
>
> (4) The logic is that to avoid memorization we want large mutual information between prediction $\hat{y}^*$ and training data D. We encourage this by upper bounding the negative part in the decomposition. Intuitively, the training objective encourages low prediction error and low mutual information between $\hat{y}^*$ and x*, which encourages the model to use the task training data D to make predictions. We added these explanations in the updated Sec. 4.1.
>
>
> Q3) It is better to give examples of use cases when it is impossible to make meta-train "mutually-exclusive" [...].
>
> A3:  We have revised the third paragraph of Sec.3 to make the patient example more specific. It now describes a scenario where it is not possible to make it mutually exclusive: the task is to recommend prescriptions based on the symptoms for each patient and the (X, Y) pair is (symptom, prescription).  Unlike the classification task, where the X (image class) can be assigned with an arbitrary label Y by random shuffling, here the X (symptom) and Y (prescription) have a highly correlated and relevant relationship that we want the model to learn, so we cannot assign random prescriptions Y to symptoms X to make the tasks mutually-exclusive.

---

> > ### Author Response · Authors · 2019-11-10
> > **Response to Reviewer #1, part2**
> >
> >
> > Q4) An experiment comparing to others in mutually-exclusive datasets would be nice to have.
> >
> > A4:  As suggested, we added a comparison on a problem with mutually-exclusive tasks: the standard 20-way, 1-shot Omniglot problem. We report results in Figure 9 (Appendix).  We find that small values of regularization coefficient β lead to slight improvements over MAML. This indicates that meta-regularization can be used in cases where it is not known a priori whether the task distribution is mutually-exclusive or not.
> >
> > Q5) Please also clarify if all methods in all experiments are hyperparameter-tuned separately.
> >
> > A5: Yes, the optimal hyperparameters for each experiment were chosen separately for each method via cross-validation (Sec. 6.2 second paragraph). We have further clarified this in Sec. A.3.2 in the updated paper.
> >
> > Q6) For future work, does recent developments in mutual information modeling help this method in any way?
> >
> > A6: This is an excellent point! Currently, we encourage the information in data D to be applied in the prediction of $\hat{y}^*$ by restricting the information from input x* and meta-parameters θ.  Alternatively, directly maximizing the mutual information $I(\hat{y}^*; D | x^*, \theta)$ in Definition 1, might be possible using recent MI estimators such as MINE. We now point this out as a promising direction for future investigation.

---

### Official Review · AnonReviewer3 · 2019-10-22
**Official Blind Review #3**

**Rating:** 6

**Review:**


Summary:

In this paper, the authors propose a new method to alleviate the effect of meta over-fitting. The designed method is based on the information-theoretic meta-regularization objective. Experiments demonstrate the effectiveness of the proposed model.

Strong Points:

+ The authors aim to alleviate the effect of meta over-fitting. In this paper, they mainly focus on alleviating the effect of brute-force memorization in the meta-training process. The problem is important in the meta-learning field.

+ The motivation for this paper is clear. The authors try to maximize the mutual information between x*, \theta and \bar{y}^*, D.

+ Experiments on both sinusoid regression, pose regression and image classification show that MR-MAML outperforms MAML and MR-CNP outperforms CNP.

Weak Points:

- My first concern is about the novelty of the proposed model. The framework and the derivations are straightforward. I think the problem is very important, however, the technical contribution may not enough to be accepted. It is better for the authors to clarify their contributions.

- It will be more helpful if the authors can describe the algorithm of the meta-testing process in Appendix A.1. In the meta-testing process, do we need to sample \theta from q(\theta|\tau)? If so, is the accuracy calculated by the averaged value of tasks with sampled weight?

- I am a little curious about the results in Table 5. The results of MAML and TAML is quite high. It would be better if the authors explain more.

After rebuttal
I think the authors' response and the revised paper address most of my concerns. I raise my score to 6.

**Experience Assessment:**

I have published one or two papers in this area.

**Review Assessment: Checking Correctness Of Derivations And Theory:**

I assessed the sensibility of the derivations and theory.

**Review Assessment: Checking Correctness Of Experiments:**

I carefully checked the experiments.

**Review Assessment: Thoroughness In Paper Reading:**

I read the paper at least twice and used my best judgement in assessing the paper.

---

> ### Author Response · Authors · 2019-11-10
> **Clarification of primary contributions and novelties**
>
> Thanks for your acknowledgement on the significance of the memorization problem and the effectiveness of our method.  We believe that our response addresses each of the major weak points raised in the review -- we would appreciate it if you could let us know whether you have any remaining reservations, or if all of your concerns have been addressed.
>
> Q1) First concern is about the novelty of the proposed model. [...] It is better for the authors to clarify their contributions.
>
> A1: Our primary contributions and novelties are:
>
> i) We are the first to identify and formalize the memorization problem in meta-learning, a previously unappreciated issue. We find that its main cause is the non-mutually-exclusive task distribution.  Furthermore, as the reviewer notes, we demonstrate that it exists in multiple meta-learning algorithms and can significantly deteriorate performance. Hence, we believe the identification and formalization of this problem to be a significant contribution.
>
> ii) We propose an effective and principled regularization approach, and it is not a trivial application of existing ideas. Firstly, as revealed in Table 2 and 3, vanilla regularization on all the model parameters does not solve the memorization problem. Knowing what parameters to be regularized is important for the meta-regularizer to be effective and the theory matters. Secondly, we identify that regularization on activations can fail to prevent the memorization problem and hypothesize why. This indicates that seemingly reasonable alternatives are insufficient. Finally, we consider the simplicity of our approach an advantage because it is then compatible with multiple meta-learning algorithms and easy to implement in practice.
>
> iii) We designed and constructed a novel non-mutually-exclusive pose regression dataset which can serve as a benchmark for future algorithms.
>
> In sum, the problem we identified and studied, the methods we proposed and the pose dataset we created are all novel. We believe this paper can bring awareness of the memorization problem when developing new meta-learning methodologies or applications. Moreover, the datasets and experiments we developed can provide a benchmark for further study of the memorization problem in meta-learning.

---

> > ### Author Response · Authors · 2019-11-10
> > **Point-by-point response, part 2**
> >
> >
> > Q2) Describe the algorithm of the meta-testing process. Do we need to sample θ from q(θ|τ)? If so, is the accuracy calculated by the averaged value of tasks with sampled weight?
> >
> > A2: Yes, for each task at meta-test time, we sample θ from q(θ|τ). The current results are obtained by sampling a single θ for each meta-test task. We appreciate the suggestion to average over multiple samples, and we find that using multiple samples of θ for a task and taking the average of the predictions can further improve performance. For example, in the sinusoid example, using 100 θ samples improves the test MSE for MR-CNP from 0.11 to 0.07 in 5-shot case and 0.09 to 0.06 in 10-shot case. We now describe the meta-test process in Algorithm 3 in the modified Appendix A.1.
> >
> > Q3)  I am a little curious about the results in Table 5. The results of MAML and TAML is quite high.
> >
> > A3: Note that the numbers in Table 5 are the *pre-update* accuracies during *meta-training* (higher does not necessarily mean better test performance). Pre-update accuracy means the accuracy obtained by the initial parameters θ before adapting to a specific task. High pre-update accuracy during meta-training can indicate that the model does not effectively adapt to the task training data, which can result in poor meta-testing predictions, as shown in Table 4.  We have updated Appendix A.5 to clarify this point, with the following paragraph:
> >
> > “In Table 5, we report the pre-update accuracy in meta-training for the non-mutually-exclusive classification experiment in Section 6.3. The pre-update accuracy is obtained by the initial parameters θ rather than the task adapted parameters φ. At meta-training time, for both MAML and MR-MAML the post-update accuracy obtained by using φ gets close to 1. High pre-update accuracy can reflect the memorization problem. For example, in 20-way 1-shot Omniglot example, the pre-update accuracy for MAML is 99.2% at the training time, which means only ~0.8% improvement in accuracy is due to adaptation, so the task training data is largely ignored. The pre-update training accuracy for MR-MAML is 5%, which means ~95% improvement in accuracy during training is due to the adaptation. This explains why in Table 4, the test accuracy of MR-MAML is much higher than that of MAML at the test-time, since the task training data is used to achieve fast adaptation.”

---

### Official Review · AnonReviewer2 · 2019-10-26
**Official Blind Review #2**

**Rating:** 8

**Review:**

This paper analyses a pitfall of current meta-learning algorithms, where the task can be inferred from the meta-training data alone, leaving the task-training data unused. Such a meta-learner would generalise well on the meta-training tasks, but will fail to generalise on new tasks at test time. This kind of overfitting is formalised as the memorization problem. This problem is implicitly resolved in current meta-learning algorithms by constructing mutually-exclusive meta-training tasks, which is not easy to construct in all scenarios. The paper introduces an information-theoretic meta-regularizer which forces information extraction from the task data (D) by restricting information flow from meta-parameters (\theta) and input (x^*). Experimental evaluation with one gradient based and one contextual meta-learning method, on non-mutually-exclusive tasks bring out the mettle of the proposed regulariser.

+ves:
+ The characterization of the memorization problem and the proposed regularizer are novel contributions.
+ The paper motivates the problem well, before proposing the methodology.
+ The paper is well-organised and the experimental setting is designed in a thoughtful manner.
+ The results are promising.

Concerns:
- The key hypothesis that the proposed meta-regularization method is based on - is that a model with memorization tends to be more complex. What is the basis for such an assumption? This is an important one for the work at the outset.

- Would the proposed regularizer help if mutually-exclusive meta-training tasks are available, as it forces the model to extract maximum information from task training data (D)? The paper does not comment on this, and this would have been useful to know.

- How much is the training overhead (in terms of time) incurred while adding the regularizer to the baseline methods (MAML and CNP)? The paper does not talk about this additional complexity.

- Evidently, the most important results in the Experiments section are the ones in Sec 6.3. However, the results do not clearly distinguish whether the meta-regularization was performed on the activations or weights here (earlier subsections do talk about this). This makes it difficult to make a conclusive inference on what aspect of the methodology actually helped here.

- There have been recent efforts that have attempted addressing overfitting in meta-learning. The paper mentions these efforts in Sec 5, and states that these have been used for existing settings where tasks are mutually exclusive. It would have been useful to include at least one of these methods in the experiments to see how the proposed regularization differs from them in practice.

Minor issues:
- The abstract says: “This causes the meta-learner to decide what should be learned from data and what must be inferred from the input.” - what is the difference between data and input?
- There are some minor typos in the work, which would benefit from a proofread. E.g: Sec 6.3 “neigbhor” -> “neighbor”

===== POST-REBUTTAL COMMENTS =========
I thank the authors for the response, the clarifications and the updated manuscript. I am happy to upgrade my rating to Accept.

The concerns regarding the ‘higher complexity of memorized models’ has been addressed convincingly in the narrative, and the visualization of weights for models with and without using the proposed regularizer makes the argument cogent.

Mentioning the number of  gradient steps used to obtain the results on mutually-exclusive meta-training tasks (Figure 9) in the narrative would help. Was the same number of steps used for MAML and MR-MAML experiments? The optimal \beta value would be very important if we want to use MR-MAML in situations where mutual-exclusivity of tasks is not known a-priori (as alluded to in the rebuttal).

I would encourage the authors to include training overhead (in terms of time) in the paper, even if it is minimal, as it would clear concerns of the reader. I would also highly encourage the authors to release the code as this would help easy reproducibility of the results.



**Experience Assessment:**

I have published one or two papers in this area.

**Review Assessment: Checking Correctness Of Derivations And Theory:**

I assessed the sensibility of the derivations and theory.

**Review Assessment: Checking Correctness Of Experiments:**

I carefully checked the experiments.

**Review Assessment: Thoroughness In Paper Reading:**

I read the paper at least twice and used my best judgement in assessing the paper.

---

> ### Author Response · Authors · 2019-11-10
> **Response to Reviewer #2**
>
> Thank you very much for your constructive comments and suggestions.  We have made all these suggested minor revisions in the updated paper. Below, we address the concerns raised in your review, which we believe we address in full. Please do let us know if you have any further concerns, or whether this adequately addresses all the issues that you raised with the paper.
>
> Q1) The basis for “a model with memorization tends to be more complex”
>
> A1: We elaborate on this intuition in Section 1 of the revised paper, which we copy here: “The model acquired when memorizing tasks is more complex than the model that results from task-specific adaptation because the memorization model is a single model that simultaneously performs well on all tasks. It needs to contain all information in its weights needed to do well on task test examples without looking at task training examples. Therefore, we expect the information content of the weights of a memorization model to be larger, and hence the model should be more complex.”
> For example, if each task is 1D regression on linearly related data with disjoint domains, the memorization model needs to learn a single piecewise linear function that simultaneously fits all tasks, whereas the adaptation model can simply fit a linear function and adapt the parameters for different tasks. To provide additional evidence for this statement, we added a visualization of the learned weights in the sinusoid regression example in Figure 6 which shows that the weights of the memorization model have more nonzero elements than the meta-regularized models.
>
> Q2) Would the proposed regularizer help if mutually-exclusive meta-training tasks are available?
>
> A2: As suggested, we added a comparison on a problem with mutually-exclusive tasks: the standard 20-way, 1-shot Omniglot problem. We report results in Figure 9 (Appendix).  We find that small values of regularization coefficient β lead to slight improvements over MAML. This indicates that meta-regularization can be used in cases where it is not known a priori whether the task distribution is mutually-exclusive or not.
>
> Q3) It would have been useful to include at least one of recent methods addressing overfitting in the experiments to see how the proposed regularization differs from them in practice.
>
> A3: We agree with the reviewer. In fact, we do compare with a recently proposed regularized meta-learner, task agnostic meta-learning (TAML, Jamal & Qi 2019), (see Table 4) and find that our method significantly outperforms TAML.
>
> Q4) How much is the training overhead (in terms of time) incurred while adding the regularizer?
>
> A4: The additional computation time incurred by the meta-regularization is minimal. For example, in the sinusoid example, for 10000 iterations, CNP takes 21.8 seconds while MR-CNP takes 23.8 seconds. For MR-MAML, the regularization does not influence the unrolled inner loop which typically dominates the computation. If MAML and MR-MAML use the same network architecture, for 1000 iterations, MAML takes 72.5 seconds while MR-MAML takes 79.7 seconds with two inner loop gradient steps in the sinusoid example.
>
> We’d like to address all the other points you raised as follows:
>
> > What is the difference between data and input in the abstract?
>
> By “data”, we mean the task training data D, and by “input” we mean the input x* of the task testing data. We clarified this in the abstract of the updated paper.
>
> > What regularization is used in Sec 6.3?
>
> We used regularization on the weights. We found that regularization on the activations does not work consistently (discussed in Section 6.2). We have made this clear in Section 6.2 and 6.3 of the updated paper.

---

### Decision · Program_Chairs · 2019-12-19

**Decision:**

Accept (Spotlight)

**Comment:**

The paper introduces the concept of overfitting in meta learning and proposes some solutions to address this problem. Overall, this is a good paper. It would be good if the authors could relate this work to meta learning approaches, which are based on hierarchical (Bayesian) modeling for learning a task embedding.

[1] Hausman et al. (ICLR 2018): Learning an Embedding Space for Transferable Robot Skills
https://openreview.net/pdf?id=rk07ZXZRb
[2] Saemundsson et al. (UAI 2018): Meta Reinforcement Learning with Latent Variable Gaussian Processes
http://auai.org/uai2018/proceedings/papers/235.pdf

---

> ### Author Response · Authors · 2020-01-20
> **Response to the meta-review**
>
> Thanks for the positive feedback!
>
> To answer your question:
> Our derivation builds upon the Bayesian meta-learning framework, but the practical algorithm is not specific to Bayesian methods and can be applied to a wide range of methods. Further, Bayesian meta-learning methods, including the referenced papers, do not solve the fundamental memorization problem that we uncover and analyze.
>
> Lastly, we'd like to point out that the form of overfitting that we analyze is not the only form of overfitting that can occur. The more obvious form of overfitting arises from overfitting the acquired adaptation procedure to the training tasks, while the version of overfitting we analyze results in no adaptation procedure at all.